# On The Scalability Of Forward Gradients, Evolution Strategies, And Control Variates

**Jake Levi**                                                                                    *jake.levi@stcatz.ox.ac.uk*
*Department of Computer Science,*
*University of Oxford*

**Seth Nabarro**                                                                                  *sdn09@ic.ac.uk*
*Dyson Robotics Lab,*
*Imperial College London*

**Mark van der Wilk**                                                                             *mark.vdwilk@cs.ox.ac.uk*
*Department of Computer Science,*
*University of Oxford*

**Reviewed on OpenReview:** *https://openreview.net/forum?id=s6g8yZimHE*

## Abstract

Stochastic gradient estimation methods such as Forward Gradients (FG) and Evolution Strategies (ES) have been proposed to overcome drawbacks of computing gradients with backpropagation (BP). However, FG and ES have large variance in high dimensions, connections between these methods have previously remained unclear, and while pure FG is guaranteed to be unbiased, proposed improvements have typically abandoned this property. We illuminate connections between FG and a popular variant of ES by proving mathematical equivalence on all quadratic objective functions. On an illustrative problem, we demonstrate theoretically how optimal convergence and learning rates scale unfavourably with intrinsic dimensionality and population size. We show that popular gradient descent techniques such as momentum and Adam do not address these fundamental scalability problems. We explore using control variates to reduce variance of FG while maintaining unbiasedness, and while we find limited success in improving over baselines, we also identify challenges that need to be overcome for these methods to scale effectively. Lastly we consider a biased method for variance reduction, and on a particular problem we show that this significantly outperforms the unbiased variance reduction methods that we consider. Assuming access to an asymptotically unbiased control variate, our results suggest that maintaining unbiasedness is not necessarily advantageous for variance reduction techniques, however we leave open the possibility that unbiasedness may be helpful when the control variate is asymptotically biased.

**Code:** https://github.com/jakelevi1996/forward_grad_public

## 1 Introduction

Computing gradients with backpropagation (BP) (Rumelhart et al., 1986), and using those gradients to adjust parameters of deep neural networks, has been the foundation of many major breakthroughs in the last decade of AI research (Silver et al., 2016; Brown et al., 2020; Jumper et al., 2021). But BP has various computational drawbacks compared to learning in biological neural networks, including backward locking (Jaderberg et al., 2017), and memory requirements that grow linearly with sequence length when learning from sequences. This has motivated alternative methods which *stochastically estimate* gradients (instead of exact computation) using populations of noise samples in order to overcome some of these computational drawbacks, including

methods based on Evolution Strategies (ES) (Salimans et al., 2017), and more recently Forward Gradients (FG) (Baydin et al., 2022). However, using stochastic estimation introduces variance into the optimisation procedure, which can slow down convergence and is increasingly problematic in high dimensions (Nesterov & Spokoiny, 2017), which inhibits scaling up to large models. Mounting evidence suggests that **variance reduction** techniques are necessary for improving the convergence rates of approximate gradient descent methods, while preserving computational advantages over exact BP.

Despite practical advances in improving ES and FG, much still remains to be explored. Connections between ES and FG remain unclear (despite conceptual similarities), whereas illuminating these connections could enable sharing ideas and accelerate progress for both methods. Theoretical understanding of these methods (Nesterov & Spokoiny, 2017) does not account for important practical considerations such as intrinsic dimensionality (Li et al., 2018; Aghajanyan et al., 2021), population size, and optimal scaling of hyperparameters. Whereas FG (as originally proposed) is guaranteed to compute **unbiased** gradient estimates (Baydin et al., 2022), even though variance may be large, existing approaches to improving convergence rates of FG have typically abandoned this property (Ren et al., 2022; Fournier et al., 2023). Unbiasedness of stochastic gradient estimates is a desirable property that *guarantees convergence* to an optimum using an appropriate learning rate schedule (Robbins & Monro, 1951; Moulines & Bach, 2011; Gower et al., 2021; Garrigos & Gower, 2023), which raises several questions. How important is maintaining unbiasedness in approximate gradient descent, especially when it is in conflict with reducing variance? To what extent can variance be reduced while maintaining unbiasedness? Can *reduced-variance unbiased* approaches to approximate gradient descent outperform biased alternatives?

**Contributions.** In §3 we show that FG and ES with antithetic sampling are mathematically equivalent (even though they are computationally quite different) when applied to any quadratic objective function, which applies to all of our subsequent theoretical and experimental results. In §4 we extend previous theoretical results (Nesterov & Spokoiny, 2017) on the convergence of approximate gradient methods to account for important practical considerations such as intrinsic dimensionality and population size, and optimal scaling of hyperparameters. In §4.4 we demonstrate empirically how these results apply to neural network optimisation. In §5 we show that simple attempts to improve the convergence of FG using momentum or Adam (Kingma, 2014) do not address the *fundamental* problems introduced by FG. In §6 we consider several different methods for using **control variates** to reduce the variance of FG while maintaining unbiasedness of gradient estimates, and although we find limited success in improving over FG, we also identify challenges that need to be overcome in order for these methods to scale beyond trivial optimisation problems. In particular, in §6.3 we model a particular type of control variate representing Synthetic Gradients (SG) (Jaderberg et al., 2017), and within this setup we show that given a sufficiently good control variate, it is better to simply use the biased expectation of the control variate, rather than pay the cost in variance of debiasing the control variate using FG. However if the control variate is not sufficiently good, then it is better to discard the control variate completely and use pure FG, because such a control variate actually *increases* variance and slows down convergence. In §7 we demonstrate how a biased FG method with access to directional derivatives and the same control variate model we considered in §6.3 can outperform alternative unbiased variance reduction methods.

**Significance.** Our theoretical results may accelerate progress in FG and ES by drawing explicit connections between the two methods, and provide intuition about scaling of convergence rates, learning rates, population size, and intrinsic dimensionality. Our empirical results raise significant questions about **the value of unbiasedness** in gradient estimation. Unbiasedness is a desirable property that *guarantees convergence* to an optimum using an appropriate learning rate schedule (Robbins & Monro, 1951; Moulines & Bach, 2011; Gower et al., 2021; Garrigos & Gower, 2023), however despite this guarantee, our results demonstrate that unbiasedness is not always advantageous, even when combined with variance reduction techniques. Our analysis suggests that this is because unbiasedness requires us to pay a cost in variance, which necessitates smaller learning rates. Ultimately this can have a more deleterious effect on convergence than abandoning unbiasedness in favour of alternative biased variance reduction techniques, which permit larger learning rates and faster convergence.

**Scope.** The theoretical and experimental results that we present in §4.3 and §4.4 establish problems with the scalability of FG and ES applied to smooth PL objective functions and neural network optimisation, respectively. Subsequently in §5, §6 and §7 we present theoretical and experimental results applied only to *spherical quadratic* objective functions. We will discuss our rationale for doing so in §8.

Regarding control variates, our analysis in §6.3 is based on an idealised approximation of a control variate that converges to the true gradient asymptotically, intended as a model of Synthetic Gradients (SG) (Jaderberg et al., 2017). Although SG has been proven to be asymptotically unbiased in deep linear models (Czarnecki et al., 2017), it remains unclear whether this can realistically hold in deep nonlinear models that would be used in practise (while maintaining computational advantages over BP). Considering scenarios in which the best possible control variates are not asymptotically unbiased may shift the balance in favour of unbiased variance reduction techniques over biased alternatives, however this direction is left to be explored in future work.

## 2 Related work

**Forward Gradients.** Consider an optimisation objective function $f : \mathbb{R}^D \to \mathbb{R}$ which maps a $D$-dimensional vector $x \in \mathbb{R}^D$ to a scalar $y = f(x)$. If $f$ is differentiable, then reverse-mode automatic differentiation (Baydin et al., 2018) can be used to compute *exact* gradients of $f$, which can be used to optimise $f$ using gradient descent. An alternative approach is to choose a "guess" of the gradient direction $\varepsilon \in \mathbb{R}^D$ (for example by sampling $\varepsilon \sim \mathcal{N}(0, I)$), compute the directional derivative $d$ along $\varepsilon$ using *forward*-mode automatic differentiation (Baydin et al., 2018), and then compute the gradient estimate $\hat{g}$ by scaling $\varepsilon$ by $d$:

$$d = \varepsilon^\top \left( \nabla_x \big[ f(x) \big] \right) \tag{1}$$

$$\hat{g} = d \, \varepsilon \tag{2}$$

$$= \varepsilon \varepsilon^\top \left( \nabla_x \big[ f(x) \big] \right) \tag{3}$$

It follows that $\hat{g}$ is an unbiased estimator of the true gradient if the condition in Equation 4 is satisfied:

$$\mathbb{E}\big[ \varepsilon \varepsilon^\top \big] = I \tag{4}$$

$$\Rightarrow \quad \mathbb{E}\big[ \hat{g} \big] = \nabla_x \big[ f(x) \big] \tag{5}$$

This approach was suggested by Nesterov & Spokoiny (2017), and evaluated empirically for optimising neural networks by Baydin et al. (2022), who named this the "Forward Gradient" (FG) method. Concurrently, this method was proposed and explored by Silver et al. (2021), who also considered more general "candidate directions" $\varepsilon$ which do not satisfy Equation 4, and therefore are no longer guaranteed to be unbiased.

Much subsequent work built on FG as presented by Baydin et al. (2022) and Silver et al. (2021), trying to improve performance by reducing variance, and typically abandoning unbiasedness. Ren et al. (2022) reduce the effective "perturbation dimension" by perturbing activations (instead of weights) and using local loss functions. Belouze (2022) proposes using the Rademacher distribution (instead of Gaussian) for sampling $\varepsilon$ and analyses the reduction in variance of gradient estimates. Fournier et al. (2023) systemtically compare performance of various combinations of gradient candidate directions (including random and deterministic guesses), gradient "targets" (including global and local loss functions), and "insertion points" (including weight perturbation and activation perturbation). Singhal et al. (2023) reduce the perturbation dimension further by projecting gradient guesses into the low dimensional subspace containing the true gradients using the known downstream weights and activations. Shukla & Shin (2023) present theoretical analysis of single-sample FG optimisation applied to quadratic objective functions, and empirical results applied to more general nonlinear optimisation problems. Bacho & Chu (2024) present a modification of Direct Feedback Alignment (DFA) (Nøkland, 2016), using FG to compute updates to the feedback weights. Flügel et al. (2024) compare different approaches for aggregating multiple FG gradient estimates on each time step. Panchal et al. (2025) compare FG against memory-efficient implementations of BP, for example using gradient checkpointing (Chen et al., 2016), and find that for comparable memory usage, BP can achieve higher accuracy, faster convergence, and lower computational cost than FG. Wang et al. (2025) focus on reducing

variance while maintaining low bias by computing *orthogonal* projections into the gradient subspace, which are computed using SVD, or approximated using Newton-Schulz Iterations (Jordan et al., 2024). Specific to RNNs, Tallec & Ollivier (2017) minimise memory consumption by using random noise vectors and forward-mode automatic differentiation (similar to FG) to compute unbiased low-rank estimates of the "sensitivity" matrix (Silver et al., 2021) used by RTRL (Williams & Zipser, 1989). Yu et al. (2024) propose a similar method using second-order Generalized Gauss-Newton updates in a random subspace.

**Evolution Strategies.** Evolution Strategies (ES) are a (diverse) class of "black-box" optimisation methods, which optimise an objective function $f(x)$ using only evaluations of $f$, without assuming access to full gradients of $f$ (as in BP) or even directional derivatives (as in FG). A popular early ES method is CMA-ES (Hansen & Ostermeier, 1996) which adapts a full covariance matrix for sampling perturbations to $x$. CMA-ES has not found widespread adoption for large-scale machine learning optimisation because of the computational cost of storing and updating a full covariance matrix, which scales quadratically in the dimensionality of $x$. Salimans et al. (2017) propose a method which has become known as OpenAI-ES (Lange, 2023), by defining a loss function $\mathcal{L}$ equal to the expectation of $f$ under the sampling distribution, and computing an unbiased Monte Carlo estimate $\hat{g}_{\text{ES}}$ of the gradient of $\mathcal{L}$ with respect to the sampling parameters $x$ using a score-function estimator (Williams, 1992):

$$\mathcal{L} = \mathbb{E}_{\theta \sim p(\theta|x)}\Big[f(\theta)\Big] \tag{6}$$

$$\frac{\partial \mathcal{L}}{\partial x} = \mathbb{E}_{\theta \sim p(\theta|x)}\left[\frac{\partial}{\partial x}\Big[\log(p(\theta|x))\Big]f(\theta)\right] \tag{7}$$

$$\hat{g}_{\text{ES}} = \frac{\partial}{\partial x}\Big[\log(p(\theta|x))\Big]f(\theta) \tag{8}$$

Typically $p(\theta|x) = \mathcal{N}(\theta|x, \sigma^2 I)$ where $\sigma$ is a fixed hyperparameter, which we can reparameterise in terms of a perturbation $\varepsilon \sim \mathcal{N}(0, I)$:

$$\hat{g}_{\text{ES}} = \frac{1}{\sigma}f(x + \sigma\varepsilon)\,\varepsilon \tag{9}$$

Tran & Luong (2022) point out that if we approximate $f(x + \sigma\varepsilon)$ with a first order Taylor expansion at $x$ and take the expectation with respect to $\varepsilon$, we recover the true gradient of $f$.

In practise, antithetic sampling (Brockhoff et al., 2010) is also used, which is equivalent to averaging over equal and opposite perturbations $\varepsilon$ and $-\varepsilon$, similar to a finite difference estimate:

$$\hat{g}_{\text{ES-AS}} = \frac{1}{2\sigma}\Big(f(x + \sigma\varepsilon) - f(x - \sigma\varepsilon)\Big)\,\varepsilon \tag{10}$$

The expressions above describe the single sample estimate, however typically this is averaged over a population of $S$ perturbations $\{\varepsilon_1, \ldots, \varepsilon_S\}$. This gradient estimator can then be used for stochastic gradient descent.

Much subsequent work has built upon OpenAI-ES to try to improve performance for optimising large neural networks, however interestingly this has gone in quite different directions to the work that has built upon FG. Maheswaranathan et al. (2019) and Liu et al. (2020) propose GES and SGES respectively, which both reduce variance of ES gradient estimates by mixing between the full perturbation distribution and a low-dimensional "guiding subspace", which is computed by orthogonalising the $k$ most recent ES gradient estimates. Tran & Luong (2022) review and experimentally evaluate both GES and SGES extensively. Most similar to our work, Tang et al. (2020) propose to reduce the variance of the ES gradient estimator using the policy gradient as a *control variate*, however their method is specific to RL problems and relies on computing policy gradients using BP. Specific to unrolled computation graphs, such as for training RNNs or hyperparameter optimisation, Vicol et al. (2021), Li et al. (2023), and Vicol (2023) present methods for dividing the computation graph into a series of "truncated unrolls", and applying more frequent ES updates after each unroll, while maintaining unbiasedness.

Unlike FG, which has seen relatively little application in research beyond FG itself, ES has seen extensive application in research, for example applied to meta-learning (Song et al., 2019), adversarial meta-RL (Lu et al., 2023), learned optimisation (Goldie et al., 2024), and LLM fine-tuning (Qiu et al., 2025).

**Synthetic Gradients.** Similar to FG and ES, the Synthetic Gradients (SG) (Jaderberg et al., 2017) method also addresses computational drawbacks of BP, but takes a very different approach to doing so. While FG and ES both rely on *random* sampling to select guesses of the gradient direction, SG learns to *deterministically* approximate the gradient as a function of activations in each layer by bootstrapping from gradient approximations in downstream layers, while avoiding forward and backward locking (Jaderberg et al., 2017). SG therefore has zero variance but in general predicts biased gradient approximations, in contrast to FG which has potentially large variance but is unbiased. Further analysis of the representations learned by SG and convergence properties is presented by Czarnecki et al. (2017).

**Control Variates.** Suppose we have an unbiased estimator $\hat{g}$ for a variable $g$ that we want to estimate, and that we also have access to a random variable $\hat{m}$ which is correlated with $\hat{g}$ and has known expectation $m$. Then $\hat{m}$ can be used as a control variate to form a new estimator $\hat{g}_{\text{CV}}$, which is still unbiased, and has lower variance than $\hat{g}$ if $\hat{g}$ and $\hat{m}$ are well correlated:

$$\hat{g}_{\text{CV}} = \hat{g} - \hat{m} + m \tag{11}$$

$$\text{Var}\big[\hat{g}_{\text{CV}}\big] = \text{Var}\big[\hat{g}\big] + \text{Var}\big[\hat{m}\big] - 2\,\text{Cov}\big[\hat{g}, \hat{m}\big] \tag{12}$$

Control variates have been used for example to reduce variance in the gradient estimates for neural networks with discretely sampled latent variables, whose gradients can't directly be estimated using the reparameterisation trick (Kingma & Welling, 2013). This includes the REBAR (Tucker et al., 2017) and RELAX (Grathwohl et al., 2017) estimators, which both learn parameters of the control variate using stochastic gradient descent, where the stochastic gradient is computed using a Monte Carlo estimate of the gradient of the variance of the control variate estimator itself.

Roeder et al. (2017) highlight a desirable property that control variates should have, referred to as the "Sticking the Landing" property, which is satisfied if the variance of the control variate estimator approaches zero as the control variate estimator approaches its target. The "Sticking the Landing" property is important because if it is not satisfied, then even after the control variate estimator has converged to its target, variance in subsequent updates can perturb the control variate estimator *away* from its target.

## 3 Equivalence Of Forward Gradients And Evolution Strategies

FG and ES are both conceptually similar (using randomly sampled perturbation directions to stochastically estimate gradients), but how similar are they mathematically? Consider a general quadratic objective function $y$ with gradient $g$:

$$y = \frac{1}{2} x^\top A x + b^\top x \tag{13}$$

$$g = Ax + b \tag{14}$$

ES *without* antithetic sampling computes a gradient estimate $\hat{g}_{\text{ES}}$ for this objective function which is unbiased, similar to FG, but contains higher order powers of $\varepsilon$ which contribute to greater variance and slower expected convergence rates. However, ES *with* antithetic sampling (Brockhoff et al., 2010) computes an unbiased gradient estimator $\hat{g}_{\text{ES-AS}}$ which is *mathematically equivalent* to the FG gradient estimator (even though FG and ES are computationally very different), for *any* ES perturbation scale $\sigma$ (proof given in Appendix A.1):

$$\hat{g}_{\text{ES}} = \frac{y}{\sigma}\,\varepsilon + \varepsilon\varepsilon^\top g + \frac{\sigma}{2}\,\varepsilon\varepsilon^\top A\varepsilon \tag{15}$$

$$\hat{g}_{\text{ES-AS}} = \varepsilon\varepsilon^\top g \tag{16}$$

This equivalence between FG and ES implies that successful strategies for *improving* one method (FG or ES) may also be successful in improving the other method, and also that *scalability problems* that apply to one method may also apply to the other method. This is important because ES is of significant practical interest to the wider ML community (Song et al., 2019; Lu et al., 2023; Goldie et al., 2024; Qiu et al., 2025). We will henceforth primarily discuss FG, however it should be noted that **all conclusions about FG applied to quadratic objective functions also apply to ES**.

This result is also notable because ES is often described intuitively as optimising a "blurred" objective function (Salimans et al., 2017; Lu et al., 2023), which might lead one to believe that ES gradient estimates themselves are innacurate. In contrast, this result highlights, as mentioned in previous work (Vicol et al., 2021) that at least for quadratic objective functions, ES gradient estimates are perfectly accurate in expectation, although they may have high variance. The accuracy with which this intuition extends to non-quadratic objective functions will depend on how well such an objective function can be locally approximated by its second order Taylor series, relative to the ES perturbation scale $\sigma$.

## 4 Scalability Of Pure Forward Gradients And Backpropagation

### 4.1 Overview

How do the performance and optimal hyperparameters of FG and ES scale with intrinsic dimensionality, extrinsic dimensionality, and population size, and how do these compare with BP? In §4.2 we will consider a class of quadratic objective functions for which we can derive *exact* expected convergence rates and optimal learning rates theoretically in closed form for both FG and BP, and compare with empirical results. In §4.3 we will consider the more general class of smooth PL objective functions, for which we derive *bounds* on the expected convergence rates of FG and BP, and show that these qualitatively match our exact theoretical results for quadratic objective functions. In §4.4 we present empirical results comparing FG and BP on neural network optimisation, which support our theoretical results from §4.2 and §4.3 on a more practically relevant objective function.

### 4.2 Quadratic Objective Functions

Consider the following objective function $y_t$, with $D$-dimensional parameter $x_t \in \mathbb{R}^D$, gradient $g_t \in \mathbb{R}^D$, and intrinsic dimensionality $\Delta \leq D$:

$$y_t = \frac{1}{2} x_t^\top A x_t \tag{17}$$

$$g_t = A x_t \tag{18}$$

$$A_{ij} = \begin{cases} 1 & i = j \leq \Delta \\ 0 & \text{otherwise} \end{cases} \tag{19}$$

Consider optimising this objective function with exact gradients (BP) and FG (or equivalently ES) with population size $S$, where $(\forall s, t)\ \varepsilon_{s,t} \sim \mathcal{N}(0, I)$:

$$x_{t+1} = \begin{cases} x_t - \alpha g_t & \text{BP} \\ x_t - \alpha \left( \frac{1}{S} \sum_{s=1}^{S} \left[ \varepsilon_{s,t} \varepsilon_{s,t}^\top g_t \right] \right) & \text{FG} \end{cases} \tag{20}$$

In both cases we can derive the expected value of the objective function as an exponential function of $t$ in terms of a convergence rate $\rho$ (proof given in Appendix A.3):

$$\mathbb{E}\left[ y_t \right] = \rho^t\, y_0 \tag{21}$$

$$\text{where} \quad \rho = \begin{cases} (1-\alpha)^2 & \text{BP} \\ (1-\alpha)^2 + \frac{\alpha^2(\Delta+1)}{S} & \text{FG} \end{cases} \tag{22}$$

The term $\frac{\alpha^2(\Delta+1)}{S}$ in the convergence rate for FG can be seen as a penalty for stochasticity, which increases with *intrinsic* dimensionality and learning rate, decreases with population size, and is *independent* of the extrinsic dimensionality $D$.

For BP, we clearly have maximum stable learning rate $\hat{\alpha} = 2$, optimal learning rate $\alpha^* = 1$, and optimal convergence rate $\rho^* = 0$, meaning we can converge to the optimal value of the objective function in a single step of gradient descent. In contrast, for FG, we can derive the maximum stable learning rate $\hat{\alpha}$, optimal

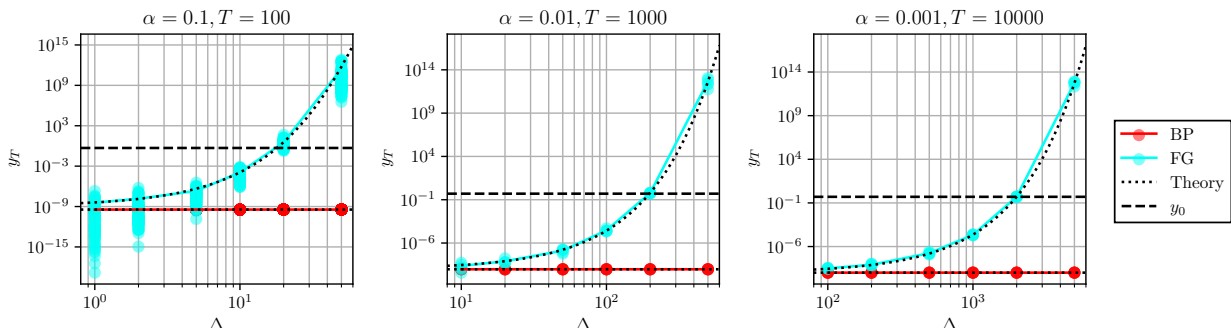

Figure 1: Performance of FG and BP applied to the $\Delta$-dimensional spherical quadratic objective function, as $\Delta$ is scaled up (with fixed learning rates). Axes are scaled logarithmically, however averages are computed in the "natural domain", consistent with our theoretical predictions.

learning rate $\alpha^*$, optimal convergence rate $\rho^*$, and minimum number of steps $T^*$ required to reach a given expected objective function value $y_T$ shown below (proof given in Appendix A.3):

$$\hat{\alpha} = \frac{2S}{S + \Delta + 1} \tag{23}$$

$$\alpha^* = \frac{S}{S + \Delta + 1} \tag{24}$$

$$\rho^* = \frac{\Delta + 1}{S + \Delta + 1} \tag{25}$$

$$T^* = \frac{\log\left(\frac{y_0}{y_T}\right)}{\log\left(1 + \frac{S}{\Delta + 1}\right)} \tag{26}$$

$$\geq \frac{\log\left(\frac{y_0}{y_T}\right)(\Delta + 1)}{S} \tag{27}$$

We see that, for FG with a fixed population size $S$, as the intrinsic dimensionality $\Delta$ increases, we must decrease the learning rate for optimal (or even just stable) performance, and that the lower bound on the number of steps $T^*$ required to reach a given level of performance scales affinely with $\Delta$. Alternatively, in order to maintain performance as $\Delta$ is scaled up, we must increase $S$ affinely with $\Delta$. With optimal learning rates, we can double $S$ in order to halve the lower bound on $T^*$, which may halve the computation time with perfect parallelisation (assuming the learning rate is scaled up appropriately). However the total FLOPS remains unchanged, because each gradient step with increased population size now requires double the number of FLOPS.

These results complement Nesterov & Spokoiny (2017) by including the optimal and maximum stable learning rates, the effect of population size, and the effect of intrinsic dimensionality (rather than just extrinsic dimensionality), which are critical to understand the success of ES applied to high-dimensional problems such as LLM fine-tuning (Qiu et al., 2025). These theoretical results show that increasing the extrinsic dimensionality beyond the intrinsic dimensionality has no effect on performance with this objective function, and that increasing population size gives performance which is equivalent to decreasing intrinsic dimensionality. Therefore, without loss of generality, we will subsequently only consider $\Delta = D$ (equivalently $A = I$, corresponding to a spherical quadratic objective function) and $S = 1$ (single-sample gradient estimates). Our theoretical results in the case $\Delta = D$ and $S = 1$ are supported by empirical results in Figure 1, demonstrating that as dimensionality is scaled up with a fixed learning rate, the performance of FG gets progressively worse and eventually becomes unstable, whereas the performance of BP does not depend on dimensionality.

Importantly, these theoretical results also show that the optimal learning rates for BP and FG are different, especially in high dimensions, which has significant implications for empirical comparisons of BP and FG.

Prior work often compares BP and FG *with the same learning rate* (Baydin et al., 2022; Ren et al., 2022), whereas for a fair comparison, future work should compare (i) BP with the learning rate which is optimal for BP against (ii) FG with the learning rate which is optimal for FG. With a low enough learning rate, our theoretical results suggest we should expect near equivalent performance for both algorithms, however this does not reflect the best performance that could be achieved with each algorithm, which is what should be compared, and may be substantially different.

### 4.3 Smooth PL Objective Functions

To what extent do the theoretical results from §4.2 apply to more general classes of objective functions? We will consider the class of objective functions which are $L$-smooth and $\mu$-PL. Smoothness is a common assumption in optimisation theory (Garrigos & Gower, 2023), and previous work has argued that PL is a good model for analysing neural network optimisation (Bassily et al., 2018; Liu et al., 2022). Consider an objective function $f : \mathbb{R}^D \to \mathbb{R}$ with gradient function $g : \mathbb{R}^D \to \mathbb{R}^D$, which is $L$-smooth (Equation 29) and $\mu$-PL (Equation 30), for all $x, u \in \mathbb{R}^D$:

$$g(x)_i = \frac{\partial f}{\partial x_i}(x) \tag{28}$$

$$\left\| g(x + u) - g(x) \right\|_2 \leq L \left\| u \right\|_2 \tag{29}$$

$$f(x) - \inf[f] \leq \frac{1}{2\mu} \left\| g(x) \right\|_2^2 \tag{30}$$

Consider optimising this objective function with learning rate $\alpha$, with exact gradients (BP) and FG with population size $S$, as described in Equation 20. In both cases, we can derive a bound on the expected convergence rate $\rho$ (proof given in Appendix A.4):

$$\mathbb{E}\left[ f(x_t) - \inf[f] \right] \leq \rho^t \left( f(x_0) - \inf[f] \right) \tag{31}$$

$$\text{where} \quad \rho = \begin{cases} 1 - 2\mu\alpha + \mu L\alpha^2 & \text{BP} \\ 1 - 2\mu\alpha + \mu L\alpha^2 \left( 1 + \frac{D+1}{S} \right) & \text{FG} \end{cases} \tag{32}$$

The bound on the convergence rate for FG is strictly weaker than the bound on the convergence rate for BP for all learning rates. Similar to §4.2, we see that for FG, the bound on the convergence rate has a penalty for stochasticity that increases with parameter dimensionality and learning rate, decreases with population size, and in this case also depends on $L$ and $\mu$. In both cases, we can find the learning rate that optimises this bound, and the resulting bound on the convergence rate:

$$\alpha^* = \begin{cases} \frac{1}{L} & \text{BP} \\ \frac{1}{L} \left( \frac{S}{S+D+1} \right) & \text{FG} \end{cases} \tag{33}$$

$$\Rightarrow \quad \rho^* = \begin{cases} 1 - \frac{\mu}{L} & \text{BP} \\ 1 - \frac{\mu}{L} \left( \frac{S}{S+D+1} \right) & \text{FG} \end{cases} \tag{34}$$

These theoretical results qualitatively match those in §4.2. As the parameter dimensionality increases, the optimal learning rate decreases and the bound on the convergence rate becomes worse, whereas as the population size increases, both the learning rate and the convergence rate bound converge to the corresponding rates for BP. In contrast, the performance of BP does not depend on dimensionality (or population size), given $L$ and $\mu$.

### 4.4 Neural Network Optimisation

How well do the theoretical results in §4.3 describe practical use cases such as neural network optimisation? In Figure 2, we present empirical results comparing FG and ES with different population sizes against BP, on the noisy and nonlinear problem of training MLPs on MNIST classification with mini-batches. Every

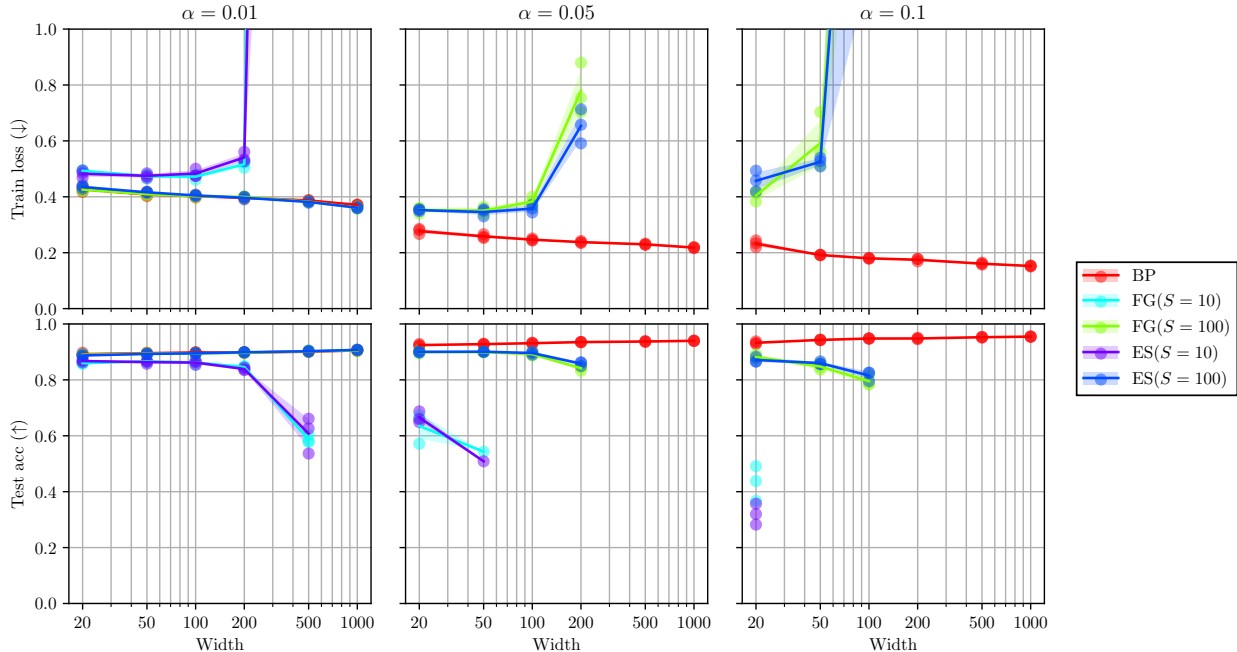

Figure 2: Final training loss (top row) and test accuracy (bottom row) of MLPs with various widths (horizontal axis) trained with BP (red), FG with different population sizes (cyan: 10, green: 100), and ES with different population sizes (purple: 10, blue: 100), and different learning rates (left: 0.01, middle: 0.05, right: 0.1) on MNIST classification. All hyperparameters are described in §4.4. Training runs in which the loss exceeded 100 were considered to have diverged, and results for these training runs are not shown.

model we train has a single hidden layer with ReLU activation function. We optimise the cross-entropy loss for 3 epochs with batch size 100 (equivalent to 1,800 total optimisation steps), and use no momentum or weight decay. For ES, we use $\sigma = 10^{-5}$. We present both the training loss (evaluated using the full training set) and the test accuracy at the end of training. On the horizontal axis we vary the width of the MLP, which directly controls the parameter dimensionality, and we compare 3 different learning rates, 2 different population sizes for both FG and ES, and repeat each experiment with 3 different random seeds.

Our empirical results qualitatively match our theoretical results. When the learning rate is small and the population size is large, FG and ES match BP very closely (left column, green and blue lines). However, increasing the learning rate or the parameter dimensionality eventually causes FG and ES to become unstable, and decreasing the population size causes FG and ES to become unstable more easily. In contrast, the performance of BP improves with increasing learning rate and width across the range of settings we consider. FG and ES do not substantially outperform BP in any setting. While in §3 we showed that ES with antithetic sampling is mathematically equivalent to FG on *quadratic* objective functions, these results show that FG and ES have very similar performance (when matched with equal population size) even on nonlinear problems.

Our results differ from those of Baydin et al. (2022), which show FG closely matching BP, or even outperforming BP in some cases, when training MLPs for MNIST classification. However, it should be noted that Baydin et al. (2022) consider much smaller learning rates, such as $2 \times 10^{-4}$ and $2 \times 10^{-5}$, and train for a much greater number of optimisation steps, such as 20,000 for their MLP results (whereas we use larger learning rates and train for only 1,800 optimisation steps). All of our theoretical and experimental results in §4.2, §4.3 and §4.4 suggest that, with a small enough learning rate, we should expect FG and BP to have very similar average performance, because the penalty for stochasticity decreases at small learning rates. However, as discussed in §4.2, in order to argue that FG is better than BP, one must compare both algorithms with their respective *optimal* learning rates, and our results show that FG becomes unstable much more quickly as the learning rate is increased, and has worse optimal performance.

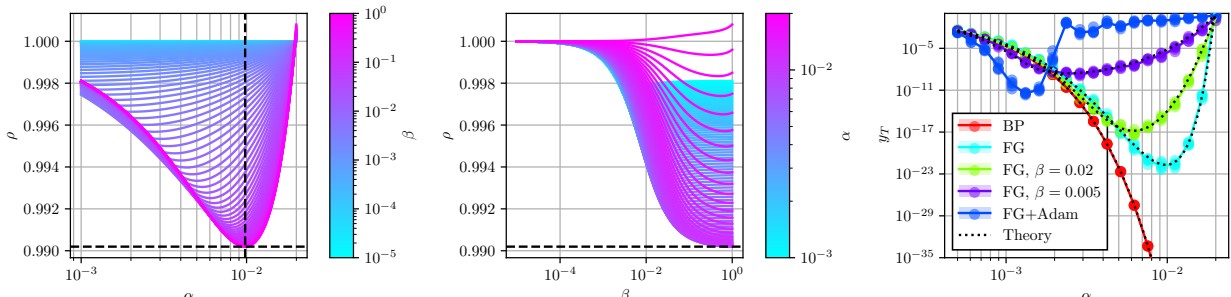

Figure 3: FG combined with alternative gradient descent schemes, applied to the spherical quadratic objective function. (Left, Middle) Convergence rate of FG with momentum averaging of the gradient estimates, as a function of $\alpha$ and $\beta$ respectively. Dashed lines show the optimal learning rate and convergence rate of pure FG (Equations 24 and 25). (Right) Performance of Adam optimisation applied to the FG gradient estimates, compared with FG (with and without momentum) and BP, as a function of learning rate, for $T = 5000$.

## 5  Momentum Is Not The Solution

Perhaps the most obvious first strategy to try to address the problem with the scalability of FG and ES would be to use a momentum average of the noisy gradient estimates. What is the purpose of momentum in gradient descent, and can it help to address the scalability problems highlighted in §4?

For non-spherical quadratic objective functions, momentum (with optimal parameters) can provably improve the convergence rate of *deterministic* gradient descent compared to the optimal learning rate without momentum, equivalent to replacing the condition number of the Hessian with its square root (Polyak, 1964; Goh, 2017). There is a common belief that momentum also improves the convergence of *stochastic* gradient descent by smoothing out noise and reducing variance in the stochastic gradients (Goodfellow, 2016; Defazio, 2020; Jelassi & Li, 2022). This may lead one to expect that using a momentum average of the noisy FG and ES gradient estimates might improve the convergence rate by reducing the variance of these gradient estimates, even for a perfectly conditioned spherical quadratic objective function. This procedure is expressed mathematically in Equations 35 and 36 for a general objective function[1]:

$$m_{t+1} = (1 - \beta)m_t + \beta \varepsilon_t \varepsilon_t^\top g_t \tag{35}$$

$$x_{t+1} = x_t - \alpha m_{t+1} \tag{36}$$

Consider applying this momentum average of the FG gradient estimates to the following spherical quadratic objective function $y_t \in \mathbb{R}$ with $D$-dimensional parameter $x_t \in \mathbb{R}^D$ and gradient $g_t \in \mathbb{R}^D$:

$$y_t = \frac{1}{2}x_t^\top x_t \tag{37}$$

$$g_t = x_t \tag{38}$$

In this the case, the asymptotic convergence rate of the objective function is equal to the spectral radius of a $3 \times 3$ matrix $A$, which can be computed in closed form. We present the expression for $A$ and its derivation in Appendix A.5. We can then compute the asymptotic convergence rate $\rho(A)$ numerically and see if there are any values $\alpha$ and $\beta$ which improve on the convergence rate $\rho^*$ of FG with optimal learning rate $\alpha^*$ without momentum (Equation 25). The results for $D = 100$ are presented in Figure 3 (Left and Middle), and demonstrate that the best convergence rate is achieved with $\beta = 1$, equivalent to not using any momentum, suggesting that momentum does not help to address the fundamental problem with the scalability of FG in this case. Analogous to §4.2, if we had access to true gradients, we could set $\beta = 1$ and $\alpha = 1$ and converge in a single step of gradient descent.

---

[1]Notably, we use a nonstandard convention for parameterising the momentum parameter $\beta$, which is equivalent to $1 - \beta$ in other work (Kingma, 2014). We choose this convention for consistency with our subsequent analyses, and for clearer and more interpretable visualisations.

Related to pure momentum is the Adam optimiser (Kingma, 2014), which combines momentum with a normalisation factor computed using second moments of the gradient estimates. Considering the ubiquitous success of Adam, a natural next approach is to try simply passing the noisy FG gradient estimates to the Adam optimiser. In Figure 3 (Right), we compare the performance of Adam with FG gradients against FG (with and without momentum) and BP as a function of learning rate, with all other Adam hyperparameters set to defaults. In these results, consistent with our results for pure momentum, Adam with the best possible learning rate actually performs worse than pure FG with optimal learning rate.

These results do not suggest that momentum or Adam won't improve performance over pure FG in more general problems, with issues such as imperfect conditioning and variable curvature. Rather, the purpose of this analysis is to *deconfound* the problems of more complex objective functions (where we already know that Adam is helpful) from the fundamental problems introduced by using noisy FG gradient estimates. The results of this analysis suggest that momentum and Adam *do not help to address these FG-specific problems*, and in fact may exacerbate them.

## 6 Unbiased Forward Gradients With Control Variates

### 6.1 Bootstrapped Synthetic Gradients

We now turn to control variates to try to improve the convergence rate of FG, by reducing variance while maintaining unbiasedness. We ideally would like a domain-agnostic control variate method which can be applied to *any* optimisation problem using FG. Perhaps the simplest approach is to learn a control variate by bootstrapping from the FG gradient estimates themselves, which are noisy but unbiased (Baydin et al., 2022), similar to how GES (Maheswaranathan et al., 2019) and SGES (Liu et al., 2020) learn a "guiding subspace" by bootstrapping from previous ES gradient estimates. Inspired by SG (Jaderberg et al., 2017), we can do this by learning a control variate $m_t$ which minimises a loss between $m_t$ and the FG gradient estimate $\hat{g}_t$. Using the squared L2 loss and gradient descent with learning rate $\beta$, we find that the control variate $m_t$ is updated effectively as a momentum estimate of the FG gradient (equivalent to Equation 35):

$$\mathcal{L}\left(m_t, \hat{g}_t\right) = \frac{1}{2}\left\|m_t - \hat{g}_t\right\|_2^2 \tag{39}$$

$$m_{t+1} = m_t - \beta \frac{\partial}{\partial m_t}\Big[\mathcal{L}\left(m_t, \hat{g}_t\right)\Big] \tag{40}$$

$$= (1-\beta)m_t + \beta\hat{g}_t \tag{41}$$

Using $m_t$ as a control variate results in the following dynamics:

$$m_{t+1} = (1-\beta)m_t + \beta\varepsilon_t\varepsilon_t^\top g_t \tag{42}$$

$$x_{t+1} = x_t - \alpha\left(\varepsilon_t\varepsilon_t^\top\left(g_t - m_t\right) + m_t\right) \tag{43}$$

Equation 43 reveals two desirable properties of this approach. Firstly, taking the expectation of Equation 43, we see that this gradient estimate still maintains unbiasedness, regardless of the value of the control variate $m_t$. Secondly, we see that if the control variate $m_t$ happens to coincide with the true gradient $g_t$, then the noise terms cancel, and the computation of $x_{t+1}$ is equivalent to a deterministic step of gradient descent with the exact gradient, which has *zero variance*. Notably, at time step $t$, we use the "stale" control variate $m_t$ instead of $m_{t+1}$, in order to avoid introducing 4th powers of $\varepsilon_t$ into the computation of $x_{t+1}$, and 8th powers into $y_{t+1}$, which would significantly increase variance. Similar to §5, for the spherical quadratic objective function defined in Equations 37 and 38, we can derive the asymptotic convergence rate of this stochastic dynamical system as the spectral radius of a $3 \times 3$ matrix $A$, which we present in Appendix A.6. We then plot the asymptotic convergence rate as a function of $\alpha$ and $\beta$ for $D = 100$ in Figure 4, and see that for this specific choice of control variate, performance with the best $\alpha$ and $\beta$ is actually worse than pure FG with optimal learning rate.

Why is performance *worse* when using a control variate, and what can we do about it? Equation 43 implies that the variance of the update for $x_{t+1}$ increases as the control variate $m_t$ gets further away from the true

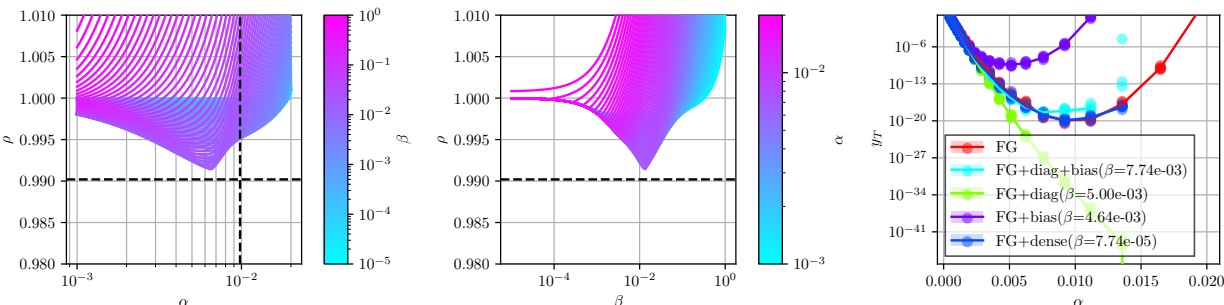

Figure 4: FG with bootstrapped control variates, applied to the spherical quadratic objective function. (Left, Middle) Performance of FG with momentum as a control variate, as a function of $\alpha$ and $\beta$ respectively. Dashed lines show the optimal learning rate and convergence rate of pure FG (Equations 24 and 25). (Right) Performance of FG with various functional control variates which have the "sticking the landing" property, as a function of $\alpha$.

gradient $g_t$. In order to converge quickly, the updates need to have low variance, which means that $m_t$ needs to converge to $g_t$ and stay there. In other words, the control variate needs to *stick the landing*. However, Equation 42 implies that even if $x_t$ was frozen and $m_t$ converged to the true gradient $g_t$, if $g_t$ is nonzero, then subsequent updates to $m_t$ will have positive variance (although they are zero in expectation) and will *perturb the control variate away* from its target. As mentioned previously, if $m_t$ does converge to $g_t$ and stay there, then the parameter updates are equivalent to deterministic gradient descent, and we have shown in §4 that this converges strictly faster than pure FG. In summary, sticking the landing is both necessary and sufficient for fast convergence of FG with a control variate, and the control variate in Equation 42 does not have this property. In the next section, we address this using "Sticking the landing" control variates.

## 6.2 Sticking The Landing

In order to develop a control variate that satisfies the "Sticking the landing" property, we take inspiration from the REBAR (Tucker et al., 2017) and RELAX (Grathwohl et al., 2017) estimators, and consider learning the control variate to minimise the variance of the control-variate-corrected gradient estimator itself. In order to allow the control variate to stick the landing even as $x_t$ is being learned (and so the true gradient $g_t$ is changing), we also let the control variate be a function of $x_t$ and learnable parameters $\phi_t$. Denoting the control variate function by $m(x, \phi)$, the control-variate-corrected gradient estimate by $u$, the gradient of the total variance (sum of element-wise variances) of $u$ by $h$, and dropping the dependence on $t$ for notational simplicity, we have:

$$u = \varepsilon \varepsilon^\top \Big( g - m(x, \phi) \Big) + m(x, \phi) \tag{44}$$

$$h = \frac{\partial}{\partial \phi} \left[ \mathbb{E} \left[ \| u - g \|_2^2 \right] \right] \tag{45}$$

We show in Appendix A.7 that $\hat{h}$ is an unbiased estimate for $h$:

$$\hat{h} = 2 \left( \varepsilon^\top \varepsilon - 1 \right) \left( m(x, \phi) - g \right)^\top \varepsilon \varepsilon^\top \frac{\partial m}{\partial \phi} \tag{46}$$

We see that $\hat{h}$ is scaled by a factor $2 \left( \varepsilon^\top \varepsilon - 1 \right)$ which in expectation grows affinely with the dimension $D$, which reflects the undesirable dependence of the variance (whose gradient we are estimating) on $D$. We further show in Appendix A.7 that we can compute a gradient vector which points in the same direction as $\hat{h}$ but without this variable scaling factor by instead differentiating the squared difference in the true and approximate *directional derivatives* in the direction $\varepsilon$:

$$\mathcal{L}_{\text{DD}} = \frac{1}{2}\left(\varepsilon^\top\left(m(x,\phi)-g\right)\right)^2 \tag{47}$$

$$\frac{\partial\mathcal{L}_{\text{DD}}}{\partial\phi} = \left(m(x,\phi)-g\right)^\top\varepsilon\varepsilon^\top\frac{\partial m}{\partial\phi} \tag{48}$$

We perform gradient descent to learn the control variate function parameters $\phi$ while simultaneously learning $x$. In Figure 4 (Right), we consider optimising the 100-dimensional spherical quadratic objective function defined in Equations 37 and 38, and compare the performance of various forms of control variate function, including (where $\odot$ denotes element-wise product, equivalent to multiplication by a diagonal matrix, and $\phi\subset\{d,b,A\}$ as appropriate):

$$m(x,\phi) = \begin{cases} d\odot x + b & \text{diag + bias} \\ d\odot x & \text{diag} \\ b & \text{bias} \\ Ax & \text{dense} \end{cases} \tag{49}$$

In all cases, we find the optimal learning rate for the control variate using a grid search over both learning rates. We see that only the "diag" control variate function manages to improve over the optimal pure FG. Adding a bias term to the control variate function ("diag + bias") leads to significantly worse performance. This cannot be due purely to increasing the number of control variate parameters to learn, because "diag + bias" uses $2D$ parameters and performs worse than "dense" which uses $D^2$ parameters.

Instead we suggest that the bad performance of "diag + bias" is caused by interfering gradient signals, because for any fixed value of $x$, there are infinitely many combinations of $d$ and $b$ for which the control variate $m(x,\phi)=d\odot x+b$ matches the true gradient $g=x$, as long as $b=x-d\odot x$. Therefore on every time step, $d$ and $b$ both receive gradient signals telling them to match the target gradient more closely, whereas over the range of all possible values of $x$, the unique optimal solution for the spherical quadratic is $b=\mathbf{0}$ and $d=\mathbf{1}$ (vectors of zeros and ones respectively). The poor convergence caused by interfering gradient signals between multiplicative ($d$) and additive ($b$) terms in $m(x,\phi)$, combined with the poor performance of using a bias term alone, implies that this specific method can only work effectively when the gradient function being learned has zero bias (so that we don't need to include a bias term in $m(x,\phi)$), corresponding to objective functions centred at zero[2]. This limitation clearly prevents generalising to objective functions of practical interest, which naturally have unknown (nonzero) minima.

### 6.3 Idealised Synthetic Gradients

Finally, having seen the challenges with trying to learn a global control variate *function*, we will instead assume that we have access to a control variate $m_t$ that converges exponentially and deterministically to the true gradient $g_t$, which we refer to as an "idealised synthetic gradient" (SG) control variate (Jaderberg et al., 2017):

$$m_{t+1} = (1-\beta)m_t + \beta g_t \tag{50}$$

$$x_{t+1} = x_t - \alpha\left(\varepsilon_t\varepsilon_t^\top\left(g_t-m_{t+1}\right)+m_{t+1}\right) \tag{51}$$

We will refer to this joint method (using SG as a control variate whose bias is removed using FG) as FG+SG. The update for this control variate (Equation 50) is deterministic, and equal to the expectation of the update for the bootstrapped control variate we considered in §6.1 (Equation 42). Having such a control variate that converges exponentially to the true gradient (which itself is a constantly moving target) is a weaker assumption than having a global control variate *function* (which we explored in §6.2). However, this may represent the best we can reasonably hope for in realistic optimisation problems with nonlinear gradients, wherein it is unrealistic to expect to be able to learn a globally accurate model of the gradient.

---

[2]To see why zero bias corresponds to objective functions centred at zero, consider the objective function $y=\frac{1}{2}x^\top Ax+b^\top x$ with gradient $g=Ax+b$ and stationary point $x^*=-A^{-1}b$. Zero bias in the gradient function ($b=0$) implies $x^*=0$.

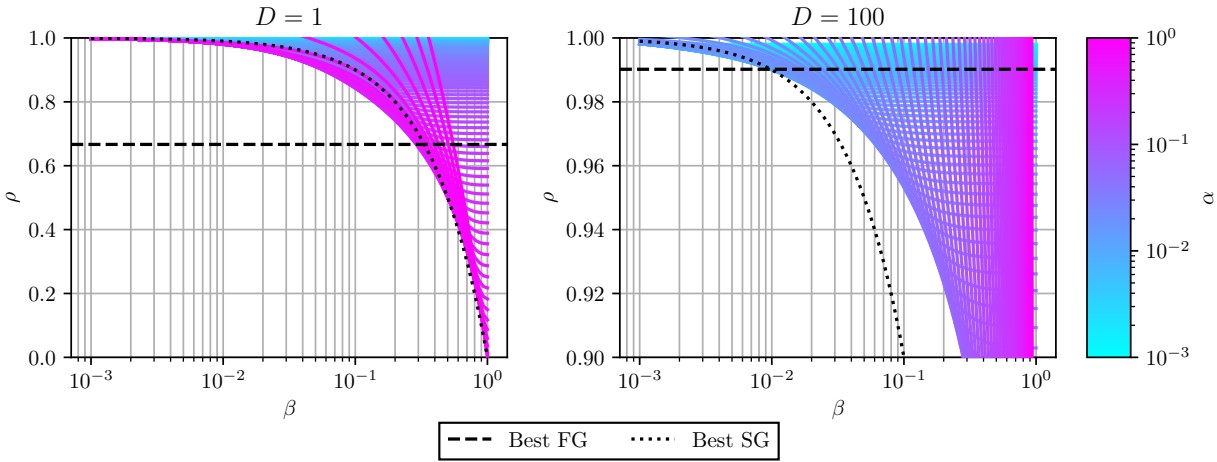

Figure 5: Convergence rate of FG+SG applied to the spherical quadratic objective function for different learning rates ($\alpha$, coloured lines) as a function of the SG convergence rate ($\beta$), compared against the best possible pure FG convergence rate (dashed line) and best possible pure SG convergence rate (dotted line), for $D = 1$ (left) and $D = 100$ (right).

The parameter $\beta$ represents how quickly the control variate $m_{t+1}$ converges to the true gradient $g_t$, with $\beta = 1$ corresponding to perfect tracking of the true gradient, in which case the dynamics of $x_t$ are equivalent to exact gradient descent. We will assume $\beta$ is fixed, for example its value might represent the best possible convergence rate of a Synthetic Gradient model (Jaderberg et al., 2017) that could be found after performing a hyperparameter sweep. We will look at various values of $\beta$, and see if there are any values for which FG+SG can outperform both (i) the best pure FG (without any control variate), and (ii) purely using the deterministic SG control variate $m_{t+1}$ as a descent direction (without any FG correction to remove bias). We will refer to the latter method as SG, corresponding to the following dynamics, which themselves are equivalent to gradient descent with momentum:

$$m_{t+1} = (1 - \beta)m_t + \beta g_t \tag{52}$$

$$x_{t+1} = x_t - \alpha m_{t+1} \tag{53}$$

As before, for the spherical quadratic objective function defined in Equations 37 and 38, we can derive the asymptotic convergence rates of both FG+SG (Equations 50 and 51) and the pure SG model (Equations 52 and 53) as the spectral radius of a $3 \times 3$ matrix, plot these convergence rates as a function of $\alpha$, $\beta$, and $D$, and compare with the convergence rate of the optimal FG (Equation 25). We present derivations and expressions of these matrices for FG+SG and pure SG in Appendices A.8 and A.9 respectively. The results for $D = 1$ and $D = 100$ are shown in Figure 5.

In both cases ($D = 1$ and $D = 100$), we see that for small $\beta$ (SG converges slowly), FG+SG outperforms SG (which demonstrates the value of debiasing synthetic gradients), but is worse than pure FG. We can understand this with reference to §6.2 (sticking the landing) because for FG+SG, $x_t$ converging depends on $m_t$ also converging, but $m_t$ is slow at sticking the landing because $\beta$ is small. For example, suppose that $x_t$ had already converged to the global minimum and has zero gradient, but $m_t$ has not yet converged. Equation 51 implies that $x_t$ will get a nonzero update from the control variate (this update is zero in expectation but has positive variance), and $x_t$ will be perturbed away from the optimum. Therefore $x_t$ cannot stick the landing until the control variate $m_t$ has also stuck the landing. Notably, in this example, a pure FG update with no control variate would leave $x_t$ exactly at the optimum. This is an example in which adding a control variate with the intention of reducing variance actually causes *increased* variance.

In both cases ($D = 1$ and $D = 100$), when $\beta$ is large (SG converges quickly), FG+SG outperforms pure FG (which demonstrates the value of reducing the variance of FG), but is worse than the best SG, especially in high dimensions. In order to understand why the optimal FG+SG is worse than the optimal pure SG

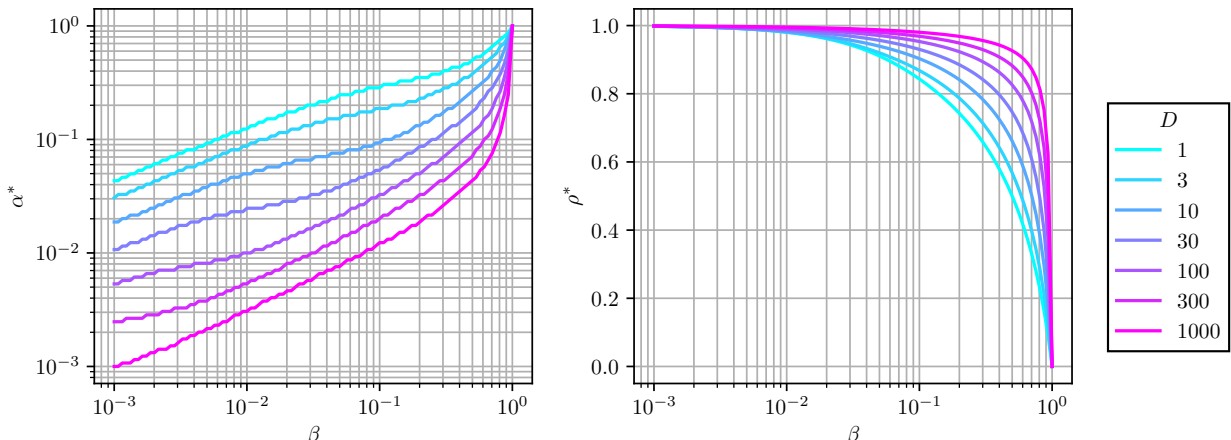

Figure 6: Optimal learning rate ($\alpha^*$, left) and convergence rate ($\rho^*$, right) for FG+SG applied to the spherical quadratic objective function as a function of the SG convergence rate ($\beta$), for various different dimensionalities ($D$, coloured lines).

in high dimensions, it is helpful to understand how the optimal learning rate $\alpha^*$ for FG+SG varies with $\beta$ and $D$, which is shown in Figure 6. Specifically, for all $\beta$, the optimal learning rate for FG+SG gets smaller in higher dimensions (consistent with the optimal learning rate for pure FG, shown in Equation 24 in §4), which reflects the increasingly deleterious effect of the *variance* added by FG in high dimensions (even though using FG *debiases* the SG). These smaller learning rates in higher dimensions are responsible for slower convergence of FG+SG compared to pure SG, for which the convergence rate and optimal learning rate are independent of dimensionality.

For FG+SG to be a useful method, we must require that for some value of $\beta$, the convergence rate is better than *both* pure FG and pure SG alone, otherwise it would be more sensible to use whichever component has the better convergence rate. In the extremely low-dimensional case ($D = 1$), we see in Figure 5 that there *is* a narrow window (between $\beta \approx 0.3$ and $\beta \approx 0.45$) in which FG+SG outperforms *both* the best pure FG and the best pure SG. This supports our original intuition that it could be advantageous to combine FG and SG into a single method which reduces the variance of FG and the bias of SG, thereby outperforming both individual components. However, scaling the dimensionality up[3] to $D = 100$, we see that this window effectively shrinks down to nothing, which we attribute to our earlier intuition, regarding the need to use progressively smaller learning rates in high dimensions to alleviate the increasingly problematic variance introduced by FG. We therefore conclude that FG+SG as presented here is not scalable, because beyond trivially small dimensionalities, the combined method is always weaker than one of its components. Therefore practitioners would benefit from using either individual method alone, depending on how quickly their SG model is able to converge (corresponding to the value of $\beta$ in Figure 5).

## 7 Biased Forward Gradients

We have shown that various attempts to reduce the variance of FG *while maintaining unbiasedness* using control variates have not been able to outperform reasonable baselines in the scenarios we have considered (except for the "diag" control variate function, whose limitations were discussed in §6.2). Prior work has proposed using FG with a biased deterministic guess of the gradient direction (Silver et al., 2021; Fournier et al., 2023), which can be expressed mathematically as follows, given some gradient guess $m$:

---

[3]Notably, $D = 100$ is still much smaller than the dimensionality of problems we would want to tackle in practise, corresponding to a single-layer neural network with only 10 input neurons and 10 output neurons, without bias.

$$d = m^\top g \tag{54}$$

$$\hat{g} = \frac{d}{m^\top m}\, m \tag{55}$$

$$= \frac{1}{m^\top m}\, m m^\top g \tag{56}$$

We will briefly analyse this method, given access to the same idealised SG that we considered in §6.3 as the gradient guess, and only directional derivatives (not exact gradients), which we refer to as Biased Forward Gradients (BFG), and which is expressed mathematically below:

$$m_{t+1} = (1 - \beta)m_t + \beta g_t \tag{57}$$

$$x_{t+1} = x_t - \alpha \left( \frac{1}{m_{t+1}^\top m_{t+1}}\, m_{t+1} m_{t+1}^\top g_t \right) \tag{58}$$

BFG computes *biased* estimates of the true gradient in general, but can it outperform the other baselines that we considered in §6.3, including those which are unbiased (FG and FG+SG)? Figure 7 shows the results of an empirical comparison between BFG, FG+SG, FG, and SG on the 100-dimensional spherical quadratic objective function defined in Equations 37 and 38, for both a slowly converging ($\beta = 0.001$) and a quickly converging ($\beta = 0.1$) SG, as a function of $\alpha$.

Firstly we note that these empirical results are largely consistent with our theoretical results in §6.3. When $\beta$ is small, FG+SG is better than SG but worse than FG, and when $\beta$ is large, FG+SG is better than FG but worse than SG. Secondly we note that, with an appropriately large learning rate, BFG outperforms all other baselines, including those which are unbiased. BFG outperforms the unbiased baselines (FG+SG and FG) because BFG can use larger learning rates without accumulating variance causing instability, and larger learning rates allow faster convergence. BFG outperforms SG because, for BFG, the multiplicative interaction between the gradient guess $m$ and the true gradient $g$ (Equation 58) allows BFG to stick the landing quickly as it approaches the optimum, using only the directional derivative, and without direct access to the true gradient. In contrast, nonzero momentum causes pure SG to overshoot, even if $x$ coincides exactly with the optimum. These results demonstrate that, although unbiasedness is desirable because it guarantees convergence to an optimum using an appropriate learning rate schedule (Robbins & Monro, 1951; Moulines & Bach, 2011; Gower et al., 2021; Garrigos & Gower, 2023), *abandoning unbiasedness* in favour of biased variance reduction (in this case using SG as a normalised gradient guess instead of a control variate) can offer substantially faster convergence.

## 8 Discussion

Our first contribution in §3 was to show that FG and ES (with antithetic sampling) are mathematically equivalent when applied to any quadratic objective function. This result is significant in the context of our findings, because it implies that all of our conclusions (except those in §4.3 and §4.4) apply not only to FG, but also to ES. However, this result also has much broader implications.

As discussed in §2, research on trying to improve FG and ES respectively has gone in quite different directions, however this connection implies that any strategy which is helpful for improving one method may indeed be helpful for improving the other method. For example, reducing the perturbation dimension for FG using activation perturbation (Ren et al., 2022; Fournier et al., 2023), and further reducing variance by projecting perturbations into the gradient subspace (Singhal et al., 2023; Wang et al., 2025) may also be applicable to reducing variance of ES gradient estimates. In the context of ES, this suggestion is connected to the recent EGGROLL algorithm that reduces the ES perturbation dimension by sampling *low-rank* perturbations to weight matrices for each member in the population (Sarkar et al., 2025). However, sampling perturbations directly to *activations* (and using these to compute gradients) reduces the perturbation dimension by a factor of two compared to rank-one weight perturbations, which may further improve both convergence and the computational complexity of each update. This offers an exciting direction for future work in ES. In the

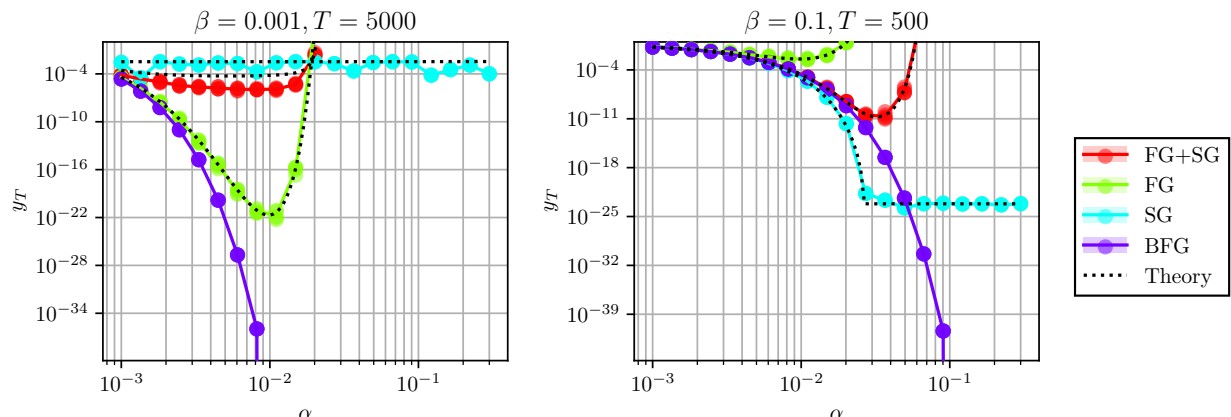

Figure 7: Comparing BFG (Equations 57 and 58) against baselines FG+SG (Equations 50 and 51), FG (Equation 20), and SG (Equations 52 and 53), applied to the spherical quadratic objective function, for a slowly converging SG (left, $\beta = 0.001$) and a quickly converging SG (right, $\beta = 0.1$). Horizontal axis shows the learning rate for the parameter being optimised.

other direction, methods for improving ES such as using guiding subspaces (Maheswaranathan et al., 2019; Liu et al., 2020; Tran & Luong, 2022) to balance exploration against exploitation in the sampling distribution may also improve performance for FG methods.

ES has been applied as a useful tool in machine learning research beyond research on ES itself (Song et al., 2019; Lu et al., 2023; Goldie et al., 2024; Qiu et al., 2025). Considering (i) the relatively limited application of FG beyond research on FG itself, (ii) the connection between FG and ES that we highlighted, and (iii) the success of ES, one might see this as a reason to be more optimistic about FG. However, it is important to note that ES has a distinct computational advantage over FG, which is that ES only requires *evaluations* of the objective function, whereas FG requires direct access to *directional derivatives* of the objective function using forward-mode automatic differentiation. This obstacle may prevent FG from being applied in many of the contexts in which ES has been successful, where evaluations are available but derivatives are not. FG does maintain the advantage over ES that its gradient estimates are guaranteed to be *unbiased* even for non-quadratic objective functions, and we will return to discuss unbiasedness shortly.

In §4.2 we considered a specific quadratic objective function (Equation 17) and derived exact convergence rates for both BP and FG, which are consistent with previous theoretical results about approximate gradient methods (Nesterov & Spokoiny, 2017). We complement these existing results by considering the effect of population size $S$ and intrinsic dimensionality $\Delta$ (not just extrinsic dimensionality $D$), and giving a clear interpretation of the penalty for stochasticity (Equation 22) and how this scales with the learning rate $\alpha$, $S$, and $\Delta$. By considering a more specific objective function than previous results, our analysis allows us to derive exact expressions for the optimal convergence rate $\rho^*$ and optimal learning rate $\alpha^*$, and their dependence on $S$ and $\Delta$. This should offer useful intuition regarding how to scale $\alpha$ and $S$ depending on $\Delta$ and the available parallelisation, and how the expected convergence rate will change as a result. In §4.3 and §4.4 we show that these results extend theoretically and empirically to smooth PL objective functions and neural network optimisation, respectively.

In §5 we show that using FG gradient estimates with more sophisticated gradient descent methods such as momentum and Adam (Kingma, 2014) which are commonly used in practise do not address the fundamental scalability problems caused by variance in the unbiased FG gradient estimates. This does not imply that momentum and Adam will make performance worse for all possible objective functions, but it does demonstrate in a simple example that they do not resolve the fundamental problems that we outlined in §4.

In §6 and §7 we attempt to address a fundamental question. Considering the guarantee that *unbiased* gradient estimates converge to an optimum using an appropriate learning rate schedule (Robbins & Monro, 1951; Moulines & Bach, 2011; Gower et al., 2021; Garrigos & Gower, 2023), can variance reduction techniques

that maintain unbiasedness improve performance more effectively than biased alternatives that abandon this guarantee? In §6 we focus specifically on using control variates for unbiased variance reduction. We consider various different approaches for acquiring control variates, including by bootstrapping from FG gradient estimates themselves, and by assuming access to a SG model (Jaderberg et al., 2017) that converges deterministically to the true gradient at some predetermined rate $\beta$. In only one case (§6.2) do we find a variance reduction method that is able to meaningfully outperform the baseline FG method, and we highlight challenges that need to be overcome in order for this method to scale to problems of practical interest. Specifically in §6.3 we show that, given access to a SG model which is used as a control variate and debiased with FG, this method can improve performance over FG *or* SG depending on $\beta$, but *not both simultaneously.* Intuitively this is because **the weaknesses of FG and SG outweigh their strengths** when they are combined into a single method using control variates.

In contrast, in §7 we consider a different *biased* method for combining FG and SG, and show that given access to the same resources, only this method reliably outperforms all other baselines (Figure 7). This does not prove that biased methods for variance reduction will always outperform unbiased methods, however it does **raise meaningful questions about the value of unbiasedness**, which is important to consider in future work trying to improve performance of FG and ES.

The control variate we considered in §6.3 and §7 was modelled on an idealised SG that converges asymptotically to the true gradient, and in this case our results suggest that unbiased variance reduction using control variates performs worse than biased alternatives. While it has been shown that asymptotic unbiasedness is a reasonable assumption for SG in deep linear models (Czarnecki et al., 2017), this may not be the case for SG in deep nonlinear models that would be used in practise. The effect of asymptotic unbiasedness of the control variate on the relative performance of unbiased variance reduction and biased alternatives may have an important bearing on whether these methods can scale up effectively to cases of practical interest. This is left as a direction for future work.

Notably, in our results we have only analysed *control variates* as a method for unbiased variance reduction. However there do exist other methods for unbiased variance reduction, such as importance sampling. This could for example be combined with guiding subspaces (Maheswaranathan et al., 2019; Liu et al., 2020; Tran & Luong, 2022) to modify the sampling distribution while maintaining unbiasedness, which is left as a direction for future work.

Besides our theoretical and experimental results in §4.3 and §4.4 applied to smooth PL objective functions and neural network optimisation respectively, our theoretical and experimental results presented in §5, §6 and §7 considered only *spherical quadratic* objective functions. Our rationale for doing so is twofold. Firstly, the spherical quadratic objective function facilitates exact theoretical analysis. Secondly, our theoretical and experimental results in §4 establish that the *fundamental problem* with the scalability of FG is present for *both* spherical quadratic objective functions *and* more complex nonlinear objective functions. This suggests that any method which *fundamentally solves* the scalability problem for FG should improve performance of FG applied to *both* spherical quadratic objective functions *and* more complex nonlinear objective functions. In this way, we consider the spherical quadratic objective function effectively as a "unit test" for evaluating whether potential modifications to FG can overcome these fundamental scalability problems. We show in §4 that pure FG does not pass this unit test, and in §5 and §6 that the momentum and control variate methods that we consider do not pass this unit test either. One surprising conclusion from these results is that this simple unit test is nontrivial without access to true gradients. We leave further development of methods which perform better at this unit test to be explored in future work, together with demonstrating that such improvements extend to problems of interest such as neural network optimisation.

Lastly, we speculate about potential motivation for applying BFG as a more general method for gradient-based optimisation, and possible advantages over conventional momentum, in Appendix B.

## Acknowledgements

JL was supported by the EPSRC Centre for Doctoral Training in Autonomous Intelligent Machines and Systems (grant number EP/S024050/1). SN was supported by EPSRC Prosperity Partnerships (EP/S036636/1) and Dyson Techonology Ltd.

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

# A  Proofs Of Theoretical Results

## A.1  Equivalence of FG and ES with antithetic sampling on quadratic objective functions

Consider the general quadratic objective function $y$ with gradient $g$ (Equations 13 and 14), the ES gradient estimator $\hat{g}_{\text{ES}}$ (Equation 9), and the ES gradient estimator with antithetic sampling $\hat{g}_{\text{ES-AS}}$ (Equation 10), restated below for convenience:

$$y = \frac{1}{2}x^\top A x + b^\top x \tag{59}$$

$$g = Ax + b \tag{60}$$

$$\hat{g}_{\text{ES}} = \frac{1}{\sigma}f(x + \sigma\varepsilon)\,\varepsilon \tag{61}$$

$$\hat{g}_{\text{ES-AS}} = \frac{1}{2\sigma}\Big(f(x + \sigma\varepsilon) - f(x - \sigma\varepsilon)\Big)\,\varepsilon \tag{62}$$

Here we assume $\varepsilon \sim \mathcal{N}(0, I)$, which implies the following identities:

$$\mathbb{E}\big[\varepsilon\big] = 0 \tag{63}$$

$$\mathbb{E}\left[\varepsilon\varepsilon^\top\right] = I \tag{64}$$

$$(\forall A) \quad \mathbb{E}\left[\varepsilon\varepsilon^\top A\varepsilon\right] = 0 \tag{65}$$

We first expand the basic ES gradient estimator for this objective function:

$$\hat{g}_{\text{ES}} = \frac{1}{\sigma}\left(\frac{1}{2}(x + \sigma\varepsilon)^\top A(x + \sigma\varepsilon) + b^\top(x + \sigma\varepsilon)\right)\varepsilon \tag{66}$$

$$= \frac{1}{\sigma}\left(\frac{1}{2}x^\top A x + \sigma\varepsilon^\top A x + \frac{\sigma^2}{2}\varepsilon^\top A\varepsilon + b^\top x + \sigma b^\top\varepsilon\right)\varepsilon \tag{67}$$

$$= \frac{1}{2\sigma}\varepsilon x^\top A x + \varepsilon\varepsilon^\top A x + \frac{\sigma}{2}\varepsilon\varepsilon^\top A\varepsilon + \frac{1}{\sigma}\varepsilon b^\top x + \varepsilon\varepsilon^\top b \tag{68}$$

$$= \frac{1}{\sigma}\left(\frac{1}{2}x^\top A x + b^\top x\right)\varepsilon + \varepsilon\varepsilon^\top\Big(Ax + b\Big) + \frac{\sigma}{2}\varepsilon\varepsilon^\top A\varepsilon \tag{69}$$

$$= \frac{y}{\sigma}\,\varepsilon + \varepsilon\varepsilon^\top g + \frac{\sigma}{2}\,\varepsilon\varepsilon^\top A\varepsilon \tag{70}$$

Taking the expectation of $\hat{g}_{\text{ES}}$, and using the identities above, we see that the basic ES gradient estimator is unbiased, although its form is quite different to the FG gradient estimator. We now expand the ES gradient estimator with antithetic sampling:

$$\hat{g}_{\text{ES-AS}} = \frac{1}{2\sigma}\Big(\ +\frac{1}{2}(x + \sigma\varepsilon)^\top A(x + \sigma\varepsilon) + b^\top(x + \sigma\varepsilon)$$
$$-\frac{1}{2}(x - \sigma\varepsilon)^\top A(x - \sigma\varepsilon) - b^\top(x - \sigma\varepsilon)\ \Big)\,\varepsilon \tag{71}$$

$$= \frac{1}{2\sigma}\Big(\ +\frac{1}{2}x^\top A x + \sigma\varepsilon^\top A x + \frac{\sigma^2}{2}\varepsilon^\top A\varepsilon + b^\top x + \sigma b^\top\varepsilon$$
$$-\frac{1}{2}x^\top A x + \sigma\varepsilon^\top A x - \frac{\sigma^2}{2}\varepsilon^\top A\varepsilon - b^\top x + \sigma b^\top\varepsilon\ \Big)\,\varepsilon \tag{72}$$

$$= \frac{1}{2\sigma}\Big((2\sigma)\,\varepsilon^\top A x + (2\sigma)\,b^\top\varepsilon\Big)\,\varepsilon \tag{73}$$

$$= \varepsilon\varepsilon^\top\Big(Ax + b\Big) \tag{74}$$

$$= \varepsilon\varepsilon^\top g \tag{75}$$

In this case the terms containing odd powers of $\varepsilon$ cancel, and we see that the ES gradient estimator with antithetic sampling is not only unbiased, but is exactly equal to the FG gradient estimator (Equation 3).

### A.2 Fourth moments of Gaussian noise vectors

We now prove some identities relating to fourth moments of Gaussian noise vectors, which will be useful in subsequent proofs. We use the following standard identities for a scalar Gaussian random variable $\varepsilon \sim \mathcal{N}(0,1)$ (Papoulis, 1965):

$$\mathbb{E}\left[\varepsilon^k\right] = \begin{cases} 0 & k = 1 \\ 1 & k = 2 \\ 0 & k = 3 \\ 3 & k = 4 \end{cases} \tag{76}$$

Assume now $\varepsilon \sim \mathcal{N}(0, I)$ is a $D$-dimensional Gaussian noise vector, and consider the fourth moment matrix $M_4 \in \mathbb{R}^{D \times D}$, where the expectation is with respect to $\varepsilon$:

$$M_4 = \mathbb{E}\left[\varepsilon \varepsilon^\top \varepsilon \varepsilon^\top\right] \tag{77}$$
$$= \mathbb{E}\left[\left(\varepsilon^\top \varepsilon\right) \varepsilon \varepsilon^\top\right] \tag{78}$$

Consider the $(i, j)$th element:

$$(M_4)_{i,j} = \mathbb{E}\left[\left(\varepsilon^\top \varepsilon\right) \varepsilon_i \varepsilon_j\right] \tag{79}$$
$$= \mathbb{E}\left[\sum_{k=1}^{D}\left[\varepsilon_k^2\right] \varepsilon_i \varepsilon_j\right] \tag{80}$$

In the case $i \neq j$ we have that $(M_4)_{i,j} = 0$, because every term in the sum in Equation 80 contains the expectation of at least one independent odd power of an element of $\varepsilon$, which is zero by Equation 76. Considering now the diagonal elements:

$$(M_4)_{i,i} = \mathbb{E}\left[\sum_{k=1}^{D}\left[\varepsilon_k^2\right] \varepsilon_i^2\right] \tag{81}$$

$$= \mathbb{E}\left[\left(\sum_{\substack{k=1 \\ k \neq i}}^{D}\left[\varepsilon_k^2\right] + \varepsilon_i^2\right) \varepsilon_i^2\right] \tag{82}$$

$$= \underbrace{\mathbb{E}\left[\left(\sum_{\substack{k=1 \\ k \neq i}}^{D}\left[\varepsilon_k^2\right]\right) \varepsilon_i^2\right]}_{=D-1} + \underbrace{\mathbb{E}\left[\varepsilon_i^4\right]}_{=3} \tag{83}$$

$$= D + 2 \tag{84}$$

All off-diagonal elements are equal to zero, and all diagonal elements are equal to $D + 2$, therefore:

$$\mathbb{E}\left[\left(\varepsilon^\top \varepsilon\right) \varepsilon \varepsilon^\top\right] = (D + 2) I \tag{85}$$

Consider now fourth moments of $\varepsilon$ involving the diagonal, binary, $D \times D$, rank-$\Delta$ matrix $A$ from Equation 19, restated below for convenience:

$$M_4^A = \mathbb{E}\left[\varepsilon \varepsilon^\top A \varepsilon \varepsilon^\top\right] \tag{86}$$

$$A_{ij} = \begin{cases} 1 & i = j \leq \Delta \\ 0 & \text{otherwise} \end{cases} \tag{87}$$

Following a similar procedure to before:

$$\varepsilon^\top A \varepsilon = \sum_{k=1}^{\Delta} \left[ \varepsilon_k^2 \right] \tag{88}$$

$$\Rightarrow \quad \left( \varepsilon \varepsilon^\top A \varepsilon \varepsilon^\top \right)_{i,j} = \sum_{k=1}^{\Delta} \left[ \varepsilon_k^2 \right] \varepsilon_i \varepsilon_j \tag{89}$$

$$\mathbb{E} \left[ \varepsilon \varepsilon^\top A \varepsilon \varepsilon^\top \right]_{i,j} = \begin{cases} \Delta + 2 & i = j \leq \Delta \\ \Delta & i = j > \Delta \\ 0 & i \neq j \end{cases} \tag{90}$$

$$\Rightarrow \quad \mathbb{E} \left[ \varepsilon \varepsilon^\top A \varepsilon \varepsilon^\top \right] = \Delta I + 2A \tag{91}$$

In the full-rank case $\Delta = D$ (which implies $A = I$), we recover Equation 85.

### A.3 Performance of BP and FG with low intrinsic dimensionality

Consider the objective function $y_t$ described by Equation 17, with $D$-dimensional parameter $x_t \in \mathbb{R}^D$, gradient $g_t \in \mathbb{R}^D$, and intrinsic dimensionality $\Delta \leq D$, restated below for convenience:

$$y_t = \frac{1}{2} x_t^\top A x_t \tag{92}$$

$$g_t = A x_t \tag{93}$$

$$A_{ij} = \begin{cases} 1 & i = j \leq \Delta \\ 0 & \text{otherwise} \end{cases} \tag{94}$$

Notably, the diagonal, binary, $D \times D$, rank-$\Delta$ matrix $A$ is symmetric and idempotent:

$$A^\top = A \tag{95}$$

$$A^2 = A \tag{96}$$

We initially consider optimising this objective function using the true gradient (BP):

$$x_{t+1} = x_t - \alpha g_t \tag{97}$$

$$= (I - \alpha A) x_t \tag{98}$$

$$\Rightarrow \quad y_{t+1} = \frac{1}{2} x_t^\top (I - \alpha A)^\top A (I - \alpha A) x_t \tag{99}$$

$$= (1 - \alpha)^2 y_t \tag{100}$$

$$\Rightarrow \quad y_t = \rho^t \, y_0 \tag{101}$$

$$\text{where} \quad \rho = (1 - \alpha)^2 \tag{102}$$

Consider now optimising the objective function $y_t$ using FG with population size $S$ (we drop the dependence of $\varepsilon_i$ on $t$ for notational convenience):

$$x_{t+1} = x_t - \alpha \left( \frac{1}{S} \sum_{i=1}^{S} \left[ \varepsilon_i \varepsilon_i^\top g_t \right] \right) \tag{103}$$

$$= \left( I - \alpha \frac{1}{S} \sum_{i=1}^{S} \left[ \varepsilon_i \varepsilon_i^\top \right] A \right) x_t \tag{104}$$

$$\Rightarrow \quad y_{t+1} = \frac{1}{2} x_t^\top \left( I - \alpha \frac{1}{S} \sum_{i=1}^{S} \left[ \varepsilon_i \varepsilon_i^\top \right] A \right)^\top A \left( I - \alpha \frac{1}{S} \sum_{j=1}^{S} \left[ \varepsilon_j \varepsilon_j^\top \right] A \right) x_t \tag{105}$$

$$= \frac{1}{2} x_t^\top \left( A - \frac{2\alpha}{S} A \sum_{i=1}^{S} \left[ \varepsilon_i \varepsilon_i^\top \right] A + \frac{\alpha^2}{S^2} A \sum_{i=1}^{S} \left[ \varepsilon_i \varepsilon_i^\top \right] A \sum_{j=1}^{S} \left[ \varepsilon_j \varepsilon_j^\top \right] A \right) x_t \tag{106}$$

Taking the expectation with respect to all $\{\varepsilon_1, \ldots, \varepsilon_S\}$, and using the result in Equation 91 from §A.2:

$$\mathbb{E}\big[y_{t+1}\big] = \frac{1}{2}x_t^\top \left( A - \frac{2\alpha}{S} A \, \mathbb{E}\left[\sum_{i=1}^{S}\big[\varepsilon_i\varepsilon_i^\top\big]\right] A + \frac{\alpha^2}{S^2} A \, \mathbb{E}\left[\sum_{i=1}^{S}\big[\varepsilon_i\varepsilon_i^\top\big] A \sum_{j=1}^{S}\big[\varepsilon_j\varepsilon_j^\top\big]\right] A \right) x_t \tag{107}$$

$$= \frac{1}{2}x_t^\top \left( (1-2\alpha)\, A + \frac{\alpha^2}{S^2} A \underbrace{\sum_{i=1}^{S}\left[\sum_{j=1}^{S}\big[\mathbb{E}\big[\varepsilon_i\varepsilon_i^\top A\varepsilon_j\varepsilon_j^\top\big]\big]\right]}_{=M} A \right) x_t \tag{108}$$

$$M = \sum_{i=1}^{S}\left[\sum_{j=1}^{S}\big[\mathbb{E}\big[\varepsilon_i\varepsilon_i^\top A\varepsilon_j\varepsilon_j^\top\big]\big]\right] \tag{109}$$

$$= \sum_{i=1}^{S}\left[\sum_{\substack{j=1 \\ j\neq i}}^{S}\big[\mathbb{E}\big[\varepsilon_i\varepsilon_i^\top A\varepsilon_j\varepsilon_j^\top\big]\big] + \mathbb{E}\big[\varepsilon_i\varepsilon_i^\top A\varepsilon_i\varepsilon_i^\top\big]\right] \tag{110}$$

$$= \sum_{i=1}^{S}\big[(S-1)A + \Delta I + 2A\big] \tag{111}$$

$$= S(S+1)A + S\Delta I \tag{112}$$

$$\Rightarrow \quad \mathbb{E}\big[y_{t+1}\big] = \frac{1}{2}x_t^\top \left( (1-2\alpha)\, A + \frac{\alpha^2}{S^2} A \left( S(S+1)A + S\Delta I\right) A \right) x_t \tag{113}$$

$$= \frac{1}{2}x_t^\top \left( (1-2\alpha)\, A + \frac{\alpha^2}{S}(S+1+\Delta)\, A \right) x_t \tag{114}$$

$$= \frac{1}{2}x_t^\top \left( \left(1-2\alpha + \alpha^2\left(1+\frac{\Delta+1}{S}\right)\right) A \right) x_t \tag{115}$$

$$= \rho\, y_t \tag{116}$$

$$\text{where} \quad \rho = (1-\alpha)^2 + \frac{\alpha^2(\Delta+1)}{S} \tag{117}$$

We can apply the same procedure on the previous time step to find $\mathbb{E}\big[y_t\big] = \rho\, y_{t-1}$, and inductively:

$$\mathbb{E}\big[y_t\big] = \rho^t\, y_0 \tag{118}$$

The maximum stable learning rate $\hat{\alpha}$ for FG is the largest value of $\alpha$ for which $\rho$ in Equation 117 is equal to one:

$$(1-\hat{\alpha})^2 + \frac{\hat{\alpha}^2(\Delta+1)}{S} = 1 \tag{119}$$

$$-2\hat{\alpha} + \hat{\alpha}^2 + \frac{\hat{\alpha}^2(\Delta+1)}{S} = 0 \tag{120}$$

$$\hat{\alpha}\frac{S+\Delta+1}{S} = 2 \tag{121}$$

$$\hat{\alpha} = \frac{2S}{S+\Delta+1} \tag{122}$$

The optimal learning rate $\alpha^*$ is the value of $\alpha$ which minimises $\rho$, and the optimal convergence rate $\rho^*$ is the value of $\rho$ when $\alpha = \alpha^*$:

$$\frac{\partial \rho}{\partial \alpha} = 2(\alpha - 1) + \frac{2\alpha(\Delta + 1)}{S} \tag{123}$$

$$= 0 \tag{124}$$

$$\Rightarrow \quad \frac{\alpha^*(S + \Delta + 1)}{S} = 1 \tag{125}$$

$$\alpha^* = \frac{S}{S + \Delta + 1} \tag{126}$$

$$\rho^* = (1 - \alpha^*)^2 + \frac{(\alpha^*)^2(\Delta + 1)}{S} \tag{127}$$

$$= \left(1 - \frac{S}{S + \Delta + 1}\right)^2 + \left(\frac{S}{S + \Delta + 1}\right)^2 \frac{\Delta + 1}{S} \tag{128}$$

$$= \left(\frac{\Delta + 1}{S + \Delta + 1}\right)^2 + \frac{S(\Delta + 1)}{(S + \Delta + 1)^2} \tag{129}$$

$$= \frac{(\Delta + 1)(S + \Delta + 1)}{(S + \Delta + 1)^2} \tag{130}$$

$$= \frac{\Delta + 1}{S + \Delta + 1} \tag{131}$$

We can express the minimum number of time steps $T^*$ required to reach an expected objective function value $y_T$ as follows:

$$y_T = (\rho^*)^{T^*} y_0 \tag{132}$$

$$\Rightarrow \quad T^* = \frac{\log\left(\frac{y_0}{y_T}\right)}{\log\left(\frac{1}{\rho^*}\right)} \tag{133}$$

$$= \frac{\log\left(\frac{y_0}{y_T}\right)}{\log\left(1 + \frac{S}{\Delta + 1}\right)} \tag{134}$$

We can bound $T^*$ by a rational function of $S$ and $\Delta$ using the following identity for the natural logarithm:

$$\log(x) \leq x - 1 \tag{135}$$

$$\Rightarrow \quad \frac{1}{\log(1 + x)} \geq \frac{1}{x} \tag{136}$$

$$T^* \geq \frac{\log\left(\frac{y_0}{y_T}\right)(\Delta + 1)}{S} \tag{137}$$

### A.4 Performance of BP and FG on smooth PL objective functions

Consider an objective function $f : \mathbb{R}^D \to \mathbb{R}$ with gradient function $g : \mathbb{R}^D \to \mathbb{R}^D$, which is $L$-smooth and $\mu$-PL, for all $x, u \in \mathbb{R}^D$:

$$g(x)_i = \frac{\partial f}{\partial x_i}(x) \tag{138}$$

$$\left\|g(x + u) - g(x)\right\|_2 \leq L\|u\|_2 \tag{139}$$

$$f(x) - \inf[f] \leq \frac{1}{2\mu}\|g(x)\|_2^2 \tag{140}$$

Analysis of optimising such an objective function with exact gradients (BP) is standard (Karimi et al., 2016; Garrigos & Gower, 2023), however we include the proof here for completeness, and for direct comparison with our proof for FG. First consider the change in the objective function after updating $x$ to $x + u$, using the fundamental theorem of calculus and Cauchy-Schwartz inequality:

$$f(x + u) - f(x) = \int_0^1 dt \left[ \frac{d}{dt} \left[ f(x + tu) \right] \right] \tag{141}$$

$$= \int_0^1 dt \left[ g(x + tu)^\top u \right] \tag{142}$$

$$= g(x)^\top u + \int_0^1 dt \left[ \left( g(x + tu) - g(x) \right)^\top u \right] \tag{143}$$

$$\leq g(x)^\top u + \int_0^1 dt \left[ \left\| g(x + tu) - g(x) \right\|_2 \|u\|_2 \right] \tag{144}$$

$$\leq g(x)^\top u + \int_0^1 dt \left[ \left( Lt \|u\|_2 \right) \|u\|_2 \right] \tag{145}$$

$$= g(x)^\top u + \frac{L}{2} u^\top u \tag{146}$$

Gradient descent with exact gradients (BP) corresponds to setting $u = -\alpha g(x)$, and assuming $\alpha \leq \frac{2}{L}$:

$$f\left( x - \alpha g(x) \right) - f(x) \leq -\alpha g(x)^\top g(x) + \frac{\alpha^2 L}{2} g(x)^\top g(x) \tag{147}$$

$$= -\alpha \left( 1 - \frac{\alpha L}{2} \right) \|g(x)\|_2^2 \tag{148}$$

$$\leq -\alpha \left( 1 - \frac{\alpha L}{2} \right) \left( 2\mu \left( f(x) - \inf[f] \right) \right) \tag{149}$$

$$= \left( -2\mu\alpha + \mu L \alpha^2 \right) \left( f(x) - \inf[f] \right) \tag{150}$$

Denoting $x_{t+1} = x_t - \alpha g(x_t)$:

$$f(x_{t+1}) - f(x_t) \leq \left( -2\mu\alpha + \mu L \alpha^2 \right) \left( f(x_t) - \inf[f] \right) \tag{151}$$

$$\Rightarrow \quad f(x_{t+1}) - \inf[f] \leq \left( 1 - 2\mu\alpha + \mu L \alpha^2 \right) \left( f(x_t) - \inf[f] \right) \tag{152}$$

$$= \rho \left( f(x_t) - \inf[f] \right) \tag{153}$$

$$\text{where} \quad \rho = 1 - 2\mu\alpha + \mu L \alpha^2 \tag{154}$$

$$\Rightarrow \quad f(x_t) - \inf[f] \leq \rho^t \left( f(x_0) - \inf[f] \right) \tag{155}$$

The learning rate $\alpha^*$ which maximises $\rho$ occurs when $\frac{\partial \rho}{\partial \alpha} = 0$, and we denote $\rho$ when $\alpha = \alpha^*$ by $\rho^*$:

$$\frac{\partial \rho}{\partial \alpha} = -2\mu + 2\mu L \alpha \tag{156}$$

$$= 0 \tag{157}$$

$$\Rightarrow \quad \alpha^* = \frac{1}{L} \tag{158}$$

$$\Rightarrow \quad \rho^* = 1 - 2\mu \left( \frac{1}{L} \right) + \mu L \left( \frac{1}{L} \right)^2 \tag{159}$$

$$= 1 - \frac{\mu}{L} \tag{160}$$

Optimisation with FG with population size $S$ corresponds to setting $u = -\alpha \hat{g}(x)$, where $\hat{g}(x)$ is defined in Equation 162:

$$f\left(x - \alpha \hat{g}(x)\right) - f(x) \leq -\alpha g(x)^{\top} \hat{g}(x) + \frac{\alpha^2 L}{2} \hat{g}(x)^{\top} \hat{g}(x) \tag{161}$$

$$\text{where} \quad \hat{g}(x) = \frac{1}{S} \sum_{i=1}^{S} \left[ \varepsilon_i \varepsilon_i^{\top} g(x) \right] \tag{162}$$

$$\text{and} \quad (\forall i) \; \varepsilon_i \sim \mathcal{N}(0, I) \tag{163}$$

We can derive the expected change in the objective function, given this parameter update:

$$\mathbb{E}\left[f\left(x - \alpha \hat{g}(x)\right) - f(x)\right] \leq -\alpha g(x)^{\top} \mathbb{E}\left[\hat{g}(x)\right] + \frac{\alpha^2 L}{2} \mathbb{E}\left[\hat{g}(x)^{\top} \hat{g}(x)\right] \tag{164}$$

$$\mathbb{E}\left[\hat{g}(x)\right] = \frac{1}{S} \mathbb{E}\left[\sum_{i=1}^{S} \left[\varepsilon_i \varepsilon_i^{\top}\right]\right] g(x) \tag{165}$$

$$= g(x) \tag{166}$$

$$\mathbb{E}\left[\hat{g}(x)^{\top} \hat{g}(x)\right] = \frac{1}{S^2} \, g(x)^{\top} \, \mathbb{E}\left[\sum_{i=1}^{S} \left[\varepsilon_i \varepsilon_i^{\top}\right] \sum_{j=1}^{S} \left[\varepsilon_j \varepsilon_j^{\top}\right]\right] g(x) \tag{167}$$

$$= \frac{1}{S^2} \, g(x)^{\top} \, \mathbb{E}\left[\sum_{i} \left[\varepsilon_i \varepsilon_i^{\top} \left(\sum_{j \neq i} \left[\varepsilon_j \varepsilon_j^{\top}\right] + \varepsilon_i \varepsilon_i^{\top}\right)\right]\right] g(x) \tag{168}$$

$$= \frac{1}{S^2} \, g(x)^{\top} \left(S(S-1) \, I + S(D+2) \, I\right) g(x) \tag{169}$$

$$= \frac{S + D + 1}{S} \, \|g(x)\|_2^2 \tag{170}$$

$$\Rightarrow \quad \mathbb{E}\left[f\left(x - \alpha \hat{g}(x)\right) - f(x)\right] \leq -\alpha \|g(x)\|_2^2 + \frac{\alpha^2 L}{2} \left(\frac{S + D + 1}{S} \|g(x)\|_2^2\right) \tag{171}$$

$$= -\alpha \left(1 - \frac{\alpha L}{2} \left(\frac{S + D + 1}{S}\right)\right) \|g(x)\|_2^2 \tag{172}$$

$$\leq -\alpha \left(1 - \frac{\alpha L}{2} \left(\frac{S + D + 1}{S}\right)\right) \left(2\mu \left(f(x) - \inf[f]\right)\right) \tag{173}$$

$$= \left(-2\mu\alpha + \mu L \alpha^2 \left(1 + \frac{D + 1}{S}\right)\right) \left(f(x) - \inf[f]\right) \tag{174}$$

Denoting $x_{t+1} = x_t - \alpha \hat{g}(x_t)$:

$$\mathbb{E}\left[f(x_{t+1}) - f(x_t)\right] \leq \left(-2\mu\alpha + \mu L \alpha^2 \left(1 + \frac{D + 1}{S}\right)\right) \left(f(x_t) - \inf[f]\right) \tag{175}$$

$$\Rightarrow \quad \mathbb{E}\left[f(x_{t+1}) - \inf[f]\right] \leq \left(1 - 2\mu\alpha + \mu L \alpha^2 \left(1 + \frac{D + 1}{S}\right)\right) \left(f(x_t) - \inf[f]\right) \tag{176}$$

$$= \rho \left(f(x_t) - \inf[f]\right) \tag{177}$$

$$\text{where} \quad \rho = 1 - 2\mu\alpha + \mu L \alpha^2 \left(1 + \frac{D + 1}{S}\right) \tag{178}$$

$$\Rightarrow \quad \mathbb{E}\left[f(x_t) - \inf[f]\right] \leq \rho^t \left(f(x_0) - \inf[f]\right) \tag{179}$$

Analogous to the case for BP, we can derive the learning rate $\alpha^*$ which optimises this bound on the expected convergence rate, and the corresponding bound on the expected convergence rate $\rho^*$:

$$\frac{\partial \rho}{\partial \alpha} = -2\mu + 2\mu L \alpha \left(\frac{S+D+1}{S}\right) \tag{180}$$

$$= 0 \tag{181}$$

$$\Rightarrow \quad \alpha^* = \frac{1}{L}\left(\frac{S}{S+D+1}\right) \tag{182}$$

$$\Rightarrow \quad \rho^* = 1 - 2\mu\left(\frac{1}{L}\left(\frac{S}{S+D+1}\right)\right) + \mu L \left(\frac{1}{L}\left(\frac{S}{S+D+1}\right)\right)^2 \left(\frac{S+D+1}{S}\right) \tag{183}$$

$$= 1 - \frac{\mu}{L}\left(\frac{S}{S+D+1}\right) \tag{184}$$

### A.5 Convergence rate of FG with momentum

Consider the spherical quadratic objective function $y_t \in \mathbb{R}$ with $D$-dimensional parameter $x_t \in \mathbb{R}^D$ and gradient $g_t \in \mathbb{R}^D$, defined in Equations 37 and 38, and restated below for convenience:

$$y_t = \frac{1}{2}x_t^\top x_t \tag{185}$$

$$g_t = x_t \tag{186}$$

Consider optimising this objective function using a momentum estimate of the FG gradient estimate, as described in Equations 35 and 36, and restated below, where $\varepsilon \sim \mathcal{N}(0, I)$, and we drop the dependence of $\varepsilon$ on $t$ for notational convenience:

$$m_{t+1} = (1-\beta)m_t + \beta\varepsilon\varepsilon^\top g_t \tag{187}$$

$$x_{t+1} = x_t - \alpha m_{t+1} \tag{188}$$

We can express $m_{t+1}$ and $x_{t+1}$ purely in terms of $m_t$, $x_t$, $\varepsilon$, $\alpha$ and $\beta$ as follows:

$$m_{t+1} = (1-\beta)\,m_t + \beta\,\varepsilon\varepsilon^\top x_t \tag{189}$$

$$x_{t+1} = \left(I - \alpha\beta\,\varepsilon\varepsilon^\top\right)x_t - \alpha(1-\beta)\,m_t \tag{190}$$

Consider now the scalar variables $z_t^{xx}$, $z_t^{xm}$, and $z_t^{mm}$ defined below, which are collected into the vector $z_t$:

$$z_t^{xx} = \mathbb{E}\left[x_t^\top x_t\right] \tag{191}$$

$$z_t^{xm} = \mathbb{E}\left[x_t^\top m_t\right] \tag{192}$$

$$z_t^{mm} = \mathbb{E}\left[m_t^\top m_t\right] \tag{193}$$

$$z_t = \begin{pmatrix} z_t^{xx} \\ z_t^{xm} \\ z_t^{mm} \end{pmatrix} \tag{194}$$

We can express $z_{t+1}$ in terms of $z_t$ as a linear recurrence relation, using results from §A.2:

$$z_{t+1}^{xx} = \mathbb{E}\left[x_{t+1}^\top x_{t+1}\right] \tag{195}$$

$$= \mathbb{E}\left[\left(\left(I - \alpha\beta\,\varepsilon\varepsilon^\top\right)x_t - \alpha(1-\beta)\,m_t\right)^\top\left(\left(I - \alpha\beta\,\varepsilon\varepsilon^\top\right)x_t - \alpha(1-\beta)\,m_t\right)\right] \tag{196}$$

$$= \mathbb{E}\Big[\ x_t^\top\left(I - \alpha\beta\,\varepsilon\varepsilon^\top\right)^\top\left(I - \alpha\beta\,\varepsilon\varepsilon^\top\right)x_t$$
$$- 2\alpha(1-\beta)\,x_t^\top\left(I - \alpha\beta\,\varepsilon\varepsilon^\top\right)^\top m_t$$
$$+ \alpha^2(1-\beta)^2\,m_t^\top m_t\ \Big] \tag{197}$$

$$= \ \left(1 - 2\alpha\beta + \alpha^2\beta^2(D+2)\right)z_t^{xx}$$
$$- 2\alpha(1-\beta)(1-\alpha\beta)\,z_t^{xm}$$
$$+ \alpha^2(1-\beta)^2\,z_t^{mm} \tag{198}$$

$$z_{t+1}^{xm} = \mathbb{E}\left[x_{t+1}^\top m_{t+1}\right] \tag{199}$$

$$= \mathbb{E}\left[\left(\left(I - \alpha\beta\,\varepsilon\varepsilon^\top\right)x_t - \alpha(1-\beta)\,m_t\right)^\top\left((1-\beta)\,m_t + \beta\,\varepsilon\varepsilon^\top x_t\right)\right] \tag{200}$$

$$= \mathbb{E}\Big[\ \beta\,x_t^\top\left(I - \alpha\beta\,\varepsilon\varepsilon^\top\right)^\top\varepsilon\varepsilon^\top x_t$$
$$+ (1-\beta)\,x_t^\top\left(I - \alpha\beta\,\varepsilon\varepsilon^\top\right)^\top m_t$$
$$- \alpha\beta(1-\beta)\,m_t^\top\varepsilon\varepsilon^\top x_t$$
$$- \alpha(1-\beta)^2\,m_t^\top m_t\ \Big] \tag{201}$$

$$= \ \beta\left(1 - \alpha\beta(D+2)\right)z_t^{xx}$$
$$+ \left((1-\beta)(1-\alpha\beta) - \alpha\beta(1-\beta)\right)z_t^{xm}$$
$$- \alpha(1-\beta)^2\,z_t^{mm} \tag{202}$$

$$z_{t+1}^{mm} = \mathbb{E}\left[m_{t+1}^\top m_{t+1}\right] \tag{203}$$

$$= \mathbb{E}\left[\left((1-\beta)\,m_t + \beta\,\varepsilon\varepsilon^\top x_t\right)^\top\left((1-\beta)\,m_t + \beta\,\varepsilon\varepsilon^\top x_t\right)\right] \tag{204}$$

$$= \mathbb{E}\Big[\ \beta^2\,x_t^\top\varepsilon\varepsilon^\top\varepsilon\varepsilon^\top x_t$$
$$+ 2\beta(1-\beta)\,x_t^\top\varepsilon\varepsilon^\top m_t$$
$$+ (1-\beta)^2\,m_t^\top m_t\ \Big] \tag{205}$$

$$= \ \beta^2(D+2)\,z_t^{xx}$$
$$+ 2\beta(1-\beta)\,z_t^{xm}$$
$$+ (1-\beta)^2\,z_t^{mm} \tag{206}$$

$$\Rightarrow \quad z_{t+1} = A z_t \tag{207}$$

$$\text{where} \quad A = \begin{pmatrix} 1 - 2\alpha\beta + \alpha^2\beta^2(D+2) & -2\alpha(1-\beta)(1-\alpha\beta) & \alpha^2(1-\beta)^2 \\ \beta\left(1 - \alpha\beta(D+2)\right) & (1-\beta)(1-2\alpha\beta) & -\alpha(1-\beta)^2 \\ \beta^2(D+2) & 2\beta(1-\beta) & (1-\beta)^2 \end{pmatrix} \tag{208}$$

$$\Rightarrow \quad z_t = A^t z_0 \tag{209}$$

It follows from Equation 209 that the asymptotic convergence rate of the components of $z_t$ is equal to the spectral radius of $A$ in Equation 208. Because $\mathbb{E}[y_t] = \frac{1}{2}z_t^{xx}$, and $z_t^{xx}$ is a component of $z_t$, it follows that the asymptotic convergence rate of $y_t$ is also equal to the spectral radius of $A$.

### A.6 Convergence rate of FG with a bootstrapped control variate

Consider the dynamics of FG with a bootstrapped control variate described in Equations 42 and 43. When applied to the spherical quadratic objective function (Equations 37 and 38), we can derive the asymptotic convergence rate, following a similar procedure to §A.5. We derive the asymptotic convergence rate below, using the same notation and method as in §A.5:

$$m_{t+1} = (1 - \beta)\, m_t + \beta\, \varepsilon\varepsilon^\top x_t \tag{210}$$

$$x_{t+1} = x_t - \alpha \left( \varepsilon\varepsilon^\top \left( x_t - m_t \right) + m_t \right) \tag{211}$$

$$= \left( I - \alpha\, \varepsilon\varepsilon^\top \right) x_t + \alpha \left( \varepsilon\varepsilon^\top - I \right) m_t \tag{212}$$

$$z_{t+1}^{xx} = \mathbb{E} \left[ x_{t+1}^\top x_{t+1} \right] \tag{213}$$

$$= \mathbb{E} \left[ \left( \left( I - \alpha\, \varepsilon\varepsilon^\top \right) x_t + \alpha \left( \varepsilon\varepsilon^\top - I \right) m_t \right)^\top \left( \left( I - \alpha\, \varepsilon\varepsilon^\top \right) x_t + \alpha \left( \varepsilon\varepsilon^\top - I \right) m_t \right) \right] \tag{214}$$

$$= \mathbb{E} \Big[ \quad x_t^\top \left( I - \alpha\, \varepsilon\varepsilon^\top \right)^\top \left( I - \alpha\, \varepsilon\varepsilon^\top \right) x_t$$
$$+ 2\alpha\, x_t^\top \left( I - \alpha\, \varepsilon\varepsilon^\top \right)^\top \left( \varepsilon\varepsilon^\top - I \right) m_t$$
$$+ \alpha^2\, m_t^\top \left( \varepsilon\varepsilon^\top - I \right)^\top \left( \varepsilon\varepsilon^\top - I \right) m_t \quad \Big] \tag{215}$$

$$= \quad \left( 1 - 2\alpha + \alpha^2(D + 2) \right) z_t^{xx}$$
$$+ 2\alpha \Big( -\alpha(D + 2) + \alpha \Big) z_t^{xm}$$
$$+ \alpha^2(D + 1)\, z_t^{mm} \tag{216}$$

$$z_{t+1}^{xm} = \mathbb{E} \left[ x_{t+1}^\top m_{t+1} \right] \tag{217}$$

$$= \mathbb{E} \left[ \left( \left( I - \alpha\, \varepsilon\varepsilon^\top \right) x_t + \alpha \left( \varepsilon\varepsilon^\top - I \right) m_t \right)^\top \left( (1 - \beta)\, m_t + \beta\, \varepsilon\varepsilon^\top x_t \right) \right] \tag{218}$$

$$= \mathbb{E} \Big[ \quad \beta\, x_t^\top \left( I - \alpha\, \varepsilon\varepsilon^\top \right)^\top \varepsilon\varepsilon^\top x_t$$
$$+ (1 - \beta)\, x_t^\top \left( I - \alpha\, \varepsilon\varepsilon^\top \right)^\top m_t$$
$$+ \alpha\beta\, m_t^\top \left( \varepsilon\varepsilon^\top - I \right)^\top \varepsilon\varepsilon^\top x_t$$
$$+ \alpha(1 - \beta)\, m_t^\top \left( \varepsilon\varepsilon^\top - I \right)^\top m_t \quad \Big] \tag{219}$$

$$= \quad \beta \Big( 1 - \alpha(D + 2) \Big) z_t^{xx}$$
$$+ \Big( (1 - \beta)(1 - \alpha) + \alpha\beta(D + 1) \Big) z_t^{xm}$$
$$+ 0\, z_t^{mm} \tag{220}$$

$$z_{t+1}^{mm} = \mathbb{E}\left[m_{t+1}^\top m_{t+1}\right] \tag{221}$$

$$= \mathbb{E}\left[\left((1-\beta)\,m_t + \beta\,\varepsilon\varepsilon^\top x_t\right)^\top\left((1-\beta)\,m_t + \beta\,\varepsilon\varepsilon^\top x_t\right)\right] \tag{222}$$

$$= \mathbb{E}\Big[\quad \beta^2\,x_t^\top\varepsilon\varepsilon^\top\varepsilon\varepsilon^\top x_t$$
$$+ 2\beta(1-\beta)\,x_t^\top\varepsilon\varepsilon^\top m_t$$
$$+ (1-\beta)^2\,m_t^\top m_t \quad\Big] \tag{223}$$

$$= \quad \beta^2(D+2)\,z_t^{xx}$$
$$+ 2\beta(1-\beta)\,z_t^{xm}$$
$$+ (1-\beta)^2\,z_t^{mm} \tag{224}$$

$$\Rightarrow \quad z_{t+1} = Az_t \tag{225}$$

$$\text{where}\quad A = \begin{pmatrix} 1 - 2\alpha + \alpha^2(D+2) & -2\alpha^2(D+1) & \alpha^2(D+1) \\ \beta\Big(1 - \alpha(D+2)\Big) & (1-\beta)(1-\alpha) + \alpha\beta(D+1) & 0 \\ \beta^2(D+2) & 2\beta(1-\beta) & (1-\beta)^2 \end{pmatrix} \tag{226}$$

$$\Rightarrow \quad z_t = A^t z_0 \tag{227}$$

Following the same reasoning in §A.5, the convergence rate of the objective function is equal to the spectral radius of the matrix $A$ in Equation 226.

### A.7 Gradient of the variance of the control-variate-corrected gradient estimator

Consider the control-variate-corrected gradient estimator $u$ and the gradient of its total variance $h$, described in Equations 44 and 45, and restated below for convenience:

$$u = \varepsilon\varepsilon^\top\Big(g - m(x,\phi)\Big) + m(x,\phi) \tag{228}$$

$$h = \frac{\partial}{\partial\phi}\left[\mathbb{E}\left[\|u - g\|_2^2\right]\right] \tag{229}$$

We can simplify $h$ as follows, noting that $\mathbb{E}[u] = g$:

$$h = \frac{\partial}{\partial \phi} \left[ \mathbb{E}\left[ (u-g)^\top (u-g) \right] \right] \tag{230}$$

$$= \frac{\partial}{\partial \phi} \left[ \mathbb{E}\left[ u^\top u - 2u^\top g + g^\top g \right] \right] \tag{231}$$

$$= \frac{\partial}{\partial \phi} \left[ \mathbb{E}\left[ u^\top u \right] - g^\top g \right] \tag{232}$$

$$= \frac{\partial}{\partial \phi} \left[ \mathbb{E}\left[ u^\top u \right] \right] - \underbrace{\frac{\partial}{\partial \phi} \left[ g^\top g \right]}_{=0} \tag{233}$$

$$= \mathbb{E}\left[ \frac{\partial}{\partial \phi} \left[ u^\top u \right] \right] \tag{234}$$

$$= \mathbb{E}\left[ 2u^\top \frac{\partial u}{\partial \phi} \right] \tag{235}$$

$$= \mathbb{E}\left[ 2u^\top \frac{\partial}{\partial \phi} \left[ \varepsilon\varepsilon^\top \left( g - m(x,\phi) \right) + m(x,\phi) \right] \right] \tag{236}$$

$$= \mathbb{E}\left[ 2u^\top \frac{\partial}{\partial \phi} \left[ \varepsilon\varepsilon^\top g + \left( I - \varepsilon\varepsilon^\top \right) m(x,\phi) \right] \right] \tag{237}$$

$$= \mathbb{E}\left[ 2u^\top \left( I - \varepsilon\varepsilon^\top \right) \frac{\partial m}{\partial \phi} \right] \tag{238}$$

$$= \mathbb{E}\left[ 2 \left( \varepsilon\varepsilon^\top \left( g - m(x,\phi) \right) + m(x,\phi) \right)^\top \left( I - \varepsilon\varepsilon^\top \right) \frac{\partial m}{\partial \phi} \right] \tag{239}$$

$$= \mathbb{E}\left[ 2 \left( g - m(x,\phi) \right)^\top \varepsilon\varepsilon^\top \left( I - \varepsilon\varepsilon^\top \right) \frac{\partial m}{\partial \phi} \right] + 2\, m(x,\phi)^\top \underbrace{\mathbb{E}\left[ I - \varepsilon\varepsilon^\top \right]}_{=0} \frac{\partial m}{\partial \phi} \tag{240}$$

$$= \mathbb{E}\left[ 2 \left( g - m(x,\phi) \right)^\top \left( \varepsilon\varepsilon^\top - \varepsilon\varepsilon^\top \varepsilon\varepsilon^\top \right) \frac{\partial m}{\partial \phi} \right] \tag{241}$$

$$= \mathbb{E}\left[ 2 \left( g - m(x,\phi) \right)^\top \left( \varepsilon\varepsilon^\top - \left( \varepsilon^\top \varepsilon \right) \varepsilon\varepsilon^\top \right) \frac{\partial m}{\partial \phi} \right] \tag{242}$$

$$= \mathbb{E}\left[ 2 \left( 1 - \varepsilon^\top \varepsilon \right) \left( g - m(x,\phi) \right)^\top \varepsilon\varepsilon^\top \frac{\partial m}{\partial \phi} \right] \tag{243}$$

$$= \mathbb{E}\left[ 2 \left( \varepsilon^\top \varepsilon - 1 \right) \left( m(x,\phi) - g \right)^\top \varepsilon\varepsilon^\top \frac{\partial m}{\partial \phi} \right] \tag{244}$$

It follows that the Monte Carlo estimator $\hat{h}$ is an unbiased estimator for $h$, where $\hat{h}$ is defined below:

$$\hat{h} = 2 \left( \varepsilon^\top \varepsilon - 1 \right) \left( m(x,\phi) - g \right)^\top \varepsilon\varepsilon^\top \frac{\partial m}{\partial \phi} \tag{245}$$

$$\Rightarrow \quad \mathbb{E}[\hat{h}] = h \tag{246}$$

Consider the squared L2 loss between the true and approximate directional derivatives in the direction $\varepsilon$, defined in Equation 47, and restated below for convenience:

$$\mathcal{L}_{\text{DD}} = \frac{1}{2} \left( \varepsilon^\top \left( m(x,\phi) - g \right) \right)^2 \tag{247}$$

We now prove that the gradient of $\mathcal{L}_{\mathrm{DD}}$ with respect to $\phi$ is a scalar multiple of $\hat{h}$:

$$\frac{\partial \mathcal{L}_{\mathrm{DD}}}{\partial \phi} = \left(\varepsilon^{\top}\left(m(x,\phi) - g\right)\right) \frac{\partial}{\partial \phi}\left[\varepsilon^{\top}\left(m(x,\phi) - g\right)\right] \tag{248}$$

$$= \left(\varepsilon^{\top}\left(m(x,\phi) - g\right)\right) \varepsilon^{\top}\frac{\partial m}{\partial \phi} \tag{249}$$

$$= \left(m(x,\phi) - g\right)\varepsilon\varepsilon^{\top}\frac{\partial m}{\partial \phi} \tag{250}$$

$$\Rightarrow \quad \hat{h} = 2\left(\varepsilon^{\top}\varepsilon - 1\right)\frac{\partial \mathcal{L}_{\mathrm{DD}}}{\partial \phi} \tag{251}$$

Interestingly, the gradient of $\mathcal{L}_{\mathrm{DD}}$ is also equal to the gradient of the loss $\mathcal{L}_{\mathrm{SG}}$ defined below, where $\perp$ denotes the "stop-gradient" operator:

$$\mathcal{L}_{\mathrm{SG}} = \frac{1}{2}\left\|m(x,\phi) - \perp\left[u\right]\right\|_2^2 \tag{252}$$

$$\frac{\partial \mathcal{L}_{\mathrm{SG}}}{\partial \phi} = \left(m(x,\phi) - u\right)^{\top}\frac{\partial}{\partial \phi}\left[m(x,\phi) - \perp\left[u\right]\right] \tag{253}$$

$$= \left(m(x,\phi) - \left(\varepsilon\varepsilon^{\top}\left(g - m(x,\phi)\right) + m(x,\phi)\right)\right)^{\top}\frac{\partial m}{\partial \phi} \tag{254}$$

$$= \left(m(x,\phi) - g\right)^{\top}\varepsilon\varepsilon^{\top}\frac{\partial m}{\partial \phi} \tag{255}$$

$$= \frac{\partial \mathcal{L}_{\mathrm{DD}}}{\partial \phi} \tag{256}$$

Removing the stop-gradient operator from the definition of $\mathcal{L}_{\mathrm{SG}}$ produces a gradient with the same direction, and magnitude scaled by $\varepsilon^{\top}\varepsilon$. The intuition of this loss function is that $m(x,\phi)$ is being updated to match its control-variate-corrected target, $u$, which also depends on $m(x,\phi)$.

### A.8 Convergence rate of FG with an idealisedised control variate

Consider the dynamics of FG+SG described in Equations 50 and 51. When applied to the spherical quadratic objective function (Equations 37 and 38), we can derive the asymptotic convergence rate, following a similar procedure to §A.5. We derive the asymptotic convergence rate below, using the same notation and method as in §A.5:

$$m_{t+1} = (1 - \beta)\, m_t + \beta\, x_t \tag{257}$$

$$x_{t+1} = x_t - \alpha\left(\varepsilon\varepsilon^{\top}\left(x_t - m_{t+1}\right) + m_{t+1}\right) \tag{258}$$

$$= \left(I - \alpha\,\varepsilon\varepsilon^{\top}\right)x_t + \alpha\left(\varepsilon\varepsilon^{\top} - I\right)\left((1 - \beta)\, m_t + \beta\, x_t\right) \tag{259}$$

$$= \left(I - \alpha\,\varepsilon\varepsilon^{\top} + \alpha\beta\left(\varepsilon\varepsilon^{\top} - I\right)\right)x_t + \alpha(1 - \beta)\left(\varepsilon\varepsilon^{\top} - I\right)m_t \tag{260}$$

$$= \left((1 - \alpha\beta)\, I - \alpha(1 - \beta)\,\varepsilon\varepsilon^{\top}\right)x_t + \alpha(1 - \beta)\left(\varepsilon\varepsilon^{\top} - I\right)m_t \tag{261}$$

$$z_{t+1}^{xx} = \mathbb{E}\left[x_{t+1}^\top x_{t+1}\right] \tag{262}$$

$$= \mathbb{E}\Big[\ \Big(\big((1-\alpha\beta)\,I - \alpha(1-\beta)\,\varepsilon\varepsilon^\top\big)x_t + \alpha(1-\beta)\left(\varepsilon\varepsilon^\top - I\right)m_t\Big)^\top$$
$$\Big(\big((1-\alpha\beta)\,I - \alpha(1-\beta)\,\varepsilon\varepsilon^\top\big)x_t + \alpha(1-\beta)\left(\varepsilon\varepsilon^\top - I\right)m_t\Big)\ \Big] \tag{263}$$

$$= \mathbb{E}\Big[\ x_t^\top\big((1-\alpha\beta)\,I - \alpha(1-\beta)\,\varepsilon\varepsilon^\top\big)^\top\big((1-\alpha\beta)\,I - \alpha(1-\beta)\,\varepsilon\varepsilon^\top\big)x_t$$
$$+ 2\alpha(1-\beta)\,x_t^\top\big((1-\alpha\beta)\,I - \alpha(1-\beta)\,\varepsilon\varepsilon^\top\big)^\top\left(\varepsilon\varepsilon^\top - I\right)m_t$$
$$+ \alpha^2(1-\beta)^2\,m_t^\top\left(\varepsilon\varepsilon^\top - I\right)^\top\left(\varepsilon\varepsilon^\top - I\right)m_t\ \Big] \tag{264}$$

$$= \ \Big((1-\alpha\beta)^2 - 2\alpha(1-\beta)(1-\alpha\beta) + \alpha^2(1-\beta)^2(D+2)\Big)\,z_t^{xx}$$
$$+ 2\alpha(1-\beta)\Big(-\alpha(1-\beta)(D+2) + \alpha(1-\beta)\Big)\,z_t^{xm}$$
$$+ \alpha^2(1-\beta)^2(D+1)\,z_t^{mm} \tag{265}$$

$$z_{t+1}^{xm} = \mathbb{E}\left[x_{t+1}^\top m_{t+1}\right] \tag{266}$$

$$= \mathbb{E}\Big[\ \Big(\big((1-\alpha\beta)\,I - \alpha(1-\beta)\,\varepsilon\varepsilon^\top\big)x_t + \alpha(1-\beta)\left(\varepsilon\varepsilon^\top - I\right)m_t\Big)^\top$$
$$\Big((1-\beta)\,m_t + \beta\,x_t\Big)\ \Big] \tag{267}$$

$$= \mathbb{E}\Big[\ \beta\,x_t^\top\big((1-\alpha\beta)\,I - \alpha(1-\beta)\,\varepsilon\varepsilon^\top\big)^\top x_t$$
$$+ (1-\beta)\,x_t^\top\big((1-\alpha\beta)\,I - \alpha(1-\beta)\,\varepsilon\varepsilon^\top\big)^\top m_t$$
$$+ \alpha\beta(1-\beta)\,m_t^\top\left(\varepsilon\varepsilon^\top - I\right)^\top x_t$$
$$+ \alpha(1-\beta)^2\,m_t^\top\left(\varepsilon\varepsilon^\top - I\right)^\top m_t\ \Big] \tag{268}$$

$$= \ \beta\Big((1-\alpha\beta) - \alpha(1-\beta)\Big)\,z_t^{xx}$$
$$+ (1-\beta)\Big((1-\alpha\beta) - \alpha(1-\beta)\Big)\,z_t^{xm}$$
$$+ 0\,z_t^{mm} \tag{269}$$

$$z_{t+1}^{mm} = \mathbb{E}\left[m_{t+1}^\top m_{t+1}\right] \tag{270}$$

$$= \mathbb{E}\left[\Big((1-\beta)\,m_t + \beta\,x_t\Big)^\top\Big((1-\beta)\,m_t + \beta\,x_t\Big)\right] \tag{271}$$

$$= \mathbb{E}\Big[\ \beta^2\,x_t^\top x_t$$
$$+ 2\beta(1-\beta)\,x_t^\top m_t$$
$$+ (1-\beta)^2\,m_t^\top m_t\ \Big] \tag{272}$$

$$= \ \beta^2\,z_t^{xx}$$
$$+ 2\beta(1-\beta)\,z_t^{xm}$$
$$+ (1-\beta)^2\,z_t^{mm} \tag{273}$$

$$\Rightarrow\quad z_{t+1} = A z_t \tag{274}$$

$$\text{where}\quad A = \begin{pmatrix} A_{xx} & -2\alpha^2(1-\beta)^2(D+1) & \alpha^2(1-\beta)^2(D+1) \\ (1-\alpha)\beta & (1-\alpha)(1-\beta) & 0 \\ \beta^2 & 2\beta(1-\beta) & (1-\beta)^2 \end{pmatrix} \tag{275}$$

$$A_{xx} = (1-\alpha\beta)^2 - 2\alpha(1-\beta)(1-\alpha\beta) + \alpha^2(1-\beta)^2(D+2) \tag{276}$$

$$\Rightarrow\quad z_t = A^t z_0 \tag{277}$$

Following the same reasoning in §A.5, the convergence rate of the objective function is equal to the spectral radius of the matrix $A$ in Equation 275.

### A.9 Convergence rate of an idealisedised synthetic gradient

Consider the dynamics of SG described in Equations 52 and 53. When applied to the spherical quadratic objective function (Equations 37 and 38), we can derive the asymptotic convergence rate, following a similar procedure to §A.5. We derive the asymptotic convergence rate below, using the same notation and method as in §A.5:

$$m_{t+1} = (1 - \beta)\, m_t + \beta\, x_t \tag{278}$$

$$x_{t+1} = x_t - \alpha\, m_{t+1} \tag{279}$$

$$= x_t - \alpha\Big((1 - \beta)\, m_t + \beta\, x_t\Big) \tag{280}$$

$$= (1 - \alpha\beta)\, x_t - \alpha(1 - \beta)\, m_t \tag{281}$$

$$z_{t+1}^{xx} = \mathbb{E}\left[x_{t+1}^\top x_{t+1}\right] \tag{282}$$

$$= \mathbb{E}\left[\Big((1 - \alpha\beta)\, x_t - \alpha(1 - \beta)\, m_t\Big)^\top \Big((1 - \alpha\beta)\, x_t - \alpha(1 - \beta)\, m_t\Big)\right] \tag{283}$$

$$= \mathbb{E}\Big[\quad (1 - \alpha\beta)^2 x_t^\top x_t$$
$$- 2\alpha(1 - \beta)(1 - \alpha\beta)\, x_t^\top m_t$$
$$+ \alpha^2(1 - \beta)^2\, m_t^\top m_t \quad \Big] \tag{284}$$

$$= \quad (1 - \alpha\beta)^2\, z_t^{xx}$$
$$- 2\alpha(1 - \beta)(1 - \alpha\beta)\, z_t^{xm}$$
$$+ \alpha^2(1 - \beta)^2\, z_t^{mm} \tag{285}$$

$$z_{t+1}^{xm} = \mathbb{E}\left[x_{t+1}^\top m_{t+1}\right] \tag{286}$$

$$= \mathbb{E}\left[\Big((1 - \alpha\beta)\, x_t - \alpha(1 - \beta)\, m_t\Big)^\top \Big((1 - \beta)\, m_t + \beta\, x_t\Big)\right] \tag{287}$$

$$= \mathbb{E}\Big[\quad \beta(1 - \alpha\beta)\, x_t^\top x_t$$
$$+ (1 - \beta)(1 - \alpha\beta)\, x_t^\top m_t$$
$$- \alpha\beta(1 - \beta)\, m_t^\top x_t$$
$$- \alpha(1 - \beta)^2\, m_t^\top m_t \quad \Big] \tag{288}$$

$$= \quad \beta(1 - \alpha\beta)\, z_t^{xx}$$
$$+ (1 - \beta)(1 - 2\alpha\beta)\, z_t^{xm}$$
$$- \alpha(1 - \beta)^2\, z_t^{mm} \tag{289}$$

$$z_{t+1}^{mm} = \mathbb{E}\left[m_{t+1}^{\top} m_{t+1}\right] \tag{290}$$

$$= \mathbb{E}\left[\left((1-\beta)\, m_t + \beta\, x_t\right)^{\top}\left((1-\beta)\, m_t + \beta\, x_t\right)\right] \tag{291}$$

$$= \mathbb{E}\Big[\quad \beta^2\, x_t^{\top} x_t$$
$$+ 2\beta(1-\beta)\, x_t^{\top} m_t$$
$$+ (1-\beta)^2\, m_t^{\top} m_t \quad\Big] \tag{292}$$

$$= \quad \beta^2\, z_t^{xx}$$
$$+ 2\beta(1-\beta)\, z_t^{xm}$$
$$+ (1-\beta)^2\, z_t^{mm} \tag{293}$$

$$\Rightarrow \quad z_{t+1} = A z_t \tag{294}$$

$$\text{where} \quad A = \begin{pmatrix} (1-\alpha\beta)^2 & -2\alpha(1-\beta)(1-\alpha\beta) & \alpha^2(1-\beta)^2 \\ \beta(1-\alpha\beta) & (1-\beta)(1-2\alpha\beta) & -\alpha(1-\beta)^2 \\ \beta^2 & 2\beta(1-\beta) & (1-\beta)^2 \end{pmatrix} \tag{295}$$

$$\Rightarrow \quad z_t = A^t z_0 \tag{296}$$

Following the same reasoning in §A.5, the convergence rate of the objective function is equal to the spectral radius of the matrix $A$ in Equation 295.

## B    Wider Significance Of Biased Forward Gradients

We briefly speculate that the BFG method we consider in §7 may have implications beyond FG and ES. Intuitively, momentum is useful for optimising badly conditioned objective functions because it smooths out oscillations in the *direction* of the gradient, that occur when using gradient descent with optimal learning rates (Goh, 2017). However, the combination of large learning rates and strong momentum prevents fast convergence because momentum causes the parameter to overshoot the optimum. This can be understood by expressing the standard momentum update as a single second-order recurrence relation (Bottou et al., 2018), which shows that the parameter update is a weighted *sum* of a steepest descent term and a momentum term:

$$\underbrace{x_{t+1} - x_t}_{\text{Parameter update}} = \underbrace{-\alpha g_t}_{\text{Steepest descent}} + \underbrace{\beta\,(x_t - x_{t-1})}_{\text{Momentum}} \tag{297}$$

This *additive* interaction between steepest descent and momentum terms prevents the magnitude of the parameter update from tending to zero as the gradient tends to zero (if the momentum is nonzero). However, the BFG method that we considered in §7 involves a *multiplicative* interaction between the momentum term and the gradient term, which ensures that the magnitude of the parameter update does tend to zero with the magnitude of the gradient. Our results in §7 showed that BFG was able to converge quickly regardless of the momentum coefficient, whereas in contrast the pure SG model (equivalent to standard momentum) was not able to converge quickly when the momentum was too strong. More generally, the multiplicative interaction in BFG may allow larger learning rates and stronger momentum to be used to smooth out oscillations in the gradient direction more effectively, while simultaneously sticking the landing and converging to the optimum more quickly without overshooting.

The BFG method may also be connected to *optimal learning rates* for minimising quadratic objective functions with arbitrary descent directions. Consider the objective function $y_t$ with gradient $g_t$, optimised using descent direction $m_t$ and learning rate $\alpha_t$:

$$y_t = \frac{1}{2} x_t^{\top} A x_t + b^{\top} x_t \tag{298}$$

$$g_t = A x_t + b \tag{299}$$

$$x_{t+1} = x_t - \alpha_t m_t \tag{300}$$

It is straightforward to derive the optimal learning rate $\alpha_t^*$ that minimises $y_{t+1}$ given $m_t$ (assuming convexity), and we see that the resulting update for $x_{t+1}$ shown in Equation 302 looks remarkably similar to the BFG update (Equation 58):

$$\alpha_t^* = \frac{m_t^\top g_t}{m_t^\top A m_t} \tag{301}$$

$$\Rightarrow \quad x_{t+1} = x_t - \frac{m_t^\top g_t}{m_t^\top A m_t} m_t \tag{302}$$

The only practical difference is the dependence of the denominator of $\alpha_t^*$ on the Hessian matrix $A$ (the BFG update effectively assumes $A = I$). Notably, for quadratic objective functions, this inner product can be computed (without computing the full Hessian matrix) using a second-order finite difference estimate requiring only three function evaluations (Glasmachers & Krause, 2020):

$$f(x) = \frac{1}{2} x^\top A x + b^\top x \tag{303}$$

$$\Rightarrow \quad m^\top A m = \frac{1}{\sigma^2} \Big( f(x + \sigma m) - 2f(x) + f(x - \sigma m) \Big) \tag{304}$$

Similar to §A.1, the directional derivative in the numerator of $\alpha_t^*$ can also be computed using two of these same function evaluations:

$$m^\top g = \frac{1}{2\sigma} \Big( f(x + \sigma m) - f(x - \sigma m) \Big) \tag{305}$$

$$\Rightarrow \quad \alpha^* = \Big( \frac{\sigma}{2} \Big) \frac{f(x + \sigma m) - f(x - \sigma m)}{f(x + \sigma m) - 2f(x) + f(x - \sigma m)} \tag{306}$$

We leave these ideas to be explored further in future work.

