# OpenReview forum: "On The Scalability Of Forward Gradients, Evolution Strategies, And Control Variates"
_TMLR — Accepted by TMLR_

### Review · Reviewer_fHyk · 2026-04-11

**Summary Of Contributions:**

This manuscript made several theoretical contributions:

1.	It proves that forward gradient (FG) is equivalent to evolution strategies (ES) with antithetic sampling when handling quadratic objective functions.

2.	It derives a convergence rate analysis with regard to intrinsic dimensionality and  populations size, providing insights in selection of optimal hyperparameters.

3.	It shows utilization of momentum or adam optimizer does not introduce siginificantly improved performance of FG.

4.	It tests several methods of control variants, but does not find significant improvements over standard FG.

5.	It argues that the value of unbiasedness may be overestimated. Theoretical analysis shows that a biased variance reduction method may significantly increase the convergence rate.

Strengths:

1.	The conclusions in the work are based on rigorous theoretical analysis.

2.	The presentations of the conclusions are clear and transparent.

3.	Challenging the necessity of maintaining unbiasedness is inspiring. This may trigger more discussions in the community.

Weakness:

1.	All the analysis and conclusions are limited to quadratic objective functions, limiting its generality to more practical setups.

2.	The focus of the paper is a bit ambiguous, with multiple standpoints to make.

3.	For most conclusions, the discussions from the ES perspective are missing.

**Audience:**

Yes

**Audience Explanation:**

This work connects FG and ES theoretically, and provides several interesting insights from the FG perspective. Both the FG and ES community should be interested to see the theoretical findings, which may further guide the development of better FG or ES variants.

**Broader Impact Concerns:**

This work is completely theoretical. No ethical concerns.

**Claims And Evidence:**

Yes

**Claims Explanation:**

All the conclusions made in this paper are supported by theoretical analysis, with clear assumptions and formulations.

**Requested Changes:**

1.	It would further broaden the scope of the work if the analysis can include more complex/noisy objective functions, or at least a discussion about how applicable are the conclusions to more practical cases. Are local optimal regions in practical use cases usually quadratic? (critical to securing my accept recommendation)

2.	A discussion about the possible limitations of the current conclusions would be helpful. (would further strengthen the work)

3.	The standard full name for ES should be “evolution” strategies, not “evolutionary” strategies. (minor issue)

---

> ### Author Response · Authors · 2026-05-01
>
> We thank the reviewer for their detailed and helpful feedback. We have made several changes to the paper to address the weaknesses and requested changes mentioned in the review, which we outline below. All figure numbers and equation numbers that we mention are referring to those in the new version of the PDF, uploaded on 30th April.
>
> - Limitations and scope of the work: this was a common criticism from all reviewers. To address this criticism, we have added two new sections and a new figure:
>   - In section 4.3, we present theoretical results for smooth PL objective functions, which generalise our earlier theoretical results. We also reference existing work that justifies the smooth PL assumption as a model for analysing neural network optimisation.
>   - In section 4.4 we present empirical results comparing FG and ES against backpropagation for training neural networks. We added a new figure (Figure 2 in the new PDF) with the numerical results for this section. We include a discussion of how these empirical results relate to our other theoretical and empirical results, and also how they relate to results from Baydin et al. (2022).
>   - In section 8, we also added a new paragraph (the penultimate paragraph, beginning "Besides our theoretical and experimental results…") explaining in more detail our rationale for considering quadratic objective functions in sections 5, 6, and 7.
> - "discussions from the ES perspective":
>   - Our intention in Section 3 is to demonstrate that FG and ES are closely related, and indeed mathematically equivalent when applied to quadratic objective functions. This implies that we don’t need to explicitly mention the ES perspective, because conclusions about ES are already implied by conclusions about FG. We have rephrased the end of the paragraph below Equation 16 to read as follows, to make this point more clear: "We will henceforth primarily discuss FG, however it should be noted that all conclusions about FG applied to quadratic objective functions also apply to ES."
>   - We have also included explicit experimental results comparing ES with FG and BP for neural network optimisation in the new Figure 2 in Section 4.4, and a discussion of the similarity between results for ES and FG in the last sentence of the penultimate paragraph of Section 4.4.
> - Correct name of Evolution Strategies: we have updated the paper, title, and abstract to correct "Evolutionary Strategies" to "Evolution Strategies".
>
> We once again thank the reviewer for their detailed and helpful review. We hope these changes and explanations address the reviewer’s concerns.

---

### Review · Reviewer_3gWT · 2026-04-15

**Summary Of Contributions:**

The paper focuses on analyzing gradient computation methods that serve as alternatives to backpropagation (reverse-mode automatic differentiation): forward gradients and evolution strategies. The authors make the following contributions:

1.	They show that FG and antithetic ES are equivalent on all quadratic functions and suggest that techniques developed for one could therefore be transferred to the other.

2.	They then analyze the convergence of ES and FG on a spherical quadratic problem and derive the convergence rate as a function of the number of samples and the intrinsic problem dimension. The result suggests that the optimal convergence rate and learning rate scale unfavorably with respect to the intrinsic dimension and population size.

3.	The authors analyze how momentum and Adam can help improve the scalability of FG and ES on the spherical quadratic problem and find evidence of only limited success.

4.	The authors then explore using unbiased control variates to reduce the variance of FG and ES. They first consider updating a control variate on the fly through gradient descent. This is equivalent to using momentum as the control variate, where the momentum is accumulated with the directional gradient estimate. They find that this performs worse than naive FG. The authors then consider learning the synthetic gradient as a parameterized function of the parameter. Here, they find limited improvement over FG, where the success is highly specific to the zero-centered quadratic assumption. Beyond this, the authors study the problem of assuming access to synthetic gradients with the “sticking the landing” property, which concretely takes the form of an exponential moving average of past gradients (momentum). Here, the authors find that FG + SG can improve either FG or SG, but not both.

5.	Finally, the authors study using real momentum as the biased direction to obtain a biased forward gradient. Here, they observe that BFG can outperform all other baselines (FG, SG, FG + SG). This demonstrates the benefit of abandoning unbiasedness for variance reduction.

Strengths:
- The paper is, in general, well written, with comprehensive related work and detailed explanations of the studied ideas.
- Because the authors focus their attention on spherical quadratic problems, they are able to compute analytical solutions that shed more light on the impact of the different algorithms.

Weaknesses:
- I think the connection between ES and FG is already largely known in the literature. For example, the MeZO paper (Malladi et al.) describes this connection in Definition 1. Vicol et al. also make use of the quadratic assumption to simplify the expression for the PES gradient estimator (Statement 4.1). I think it is acceptable to state this result formally, but I would downplay the authors’ claimed contribution on this point.
- I think the spherical quadratic setup (i.e., the isotropic Hessian assumption) is a bit too simplified. In the spherical quadratic setting, a single step of gradient descent would already minimize the loss function. In addition, in this case, momentum-style methods with access to gradients (without randomness) would simply set the momentum coefficient to \beta = 1, effectively not using momentum. In light of this, when the authors claim that momentum/Adam does not help optimize with FG gradients, it is hard to know whether this is more a property of the exact spherical quadratic setup or a genuine issue with momentum + FG. I hope the authors can further clarify their reason for sticking to the spherical quadratic assumption and whether any quantitative analysis can be done for the non-spherical case.
- I think the paper could also benefit from running actual experiments in addition to analyzing the theoretical quadratic model. It would be interesting to check whether the predictions of the theoretical model match practice.
- Minor: I think the paper could make it more explicit which exact loss function each experimental figure uses. To my understanding, they all use Equations (17)–(19), but I would like confirmation. I also think it would be useful to remind the reader what the dashed lines in Figure 3 represent.

Malladi, Sadhika, et al. "Fine-tuning language models with just forward passes." Advances in Neural Information Processing Systems 36 (2023): 53038-53075.

Vicol, Paul, Luke Metz, and Jascha Sohl-Dickstein. "Unbiased gradient estimation in unrolled computation graphs with persistent evolution strategies." International Conference on Machine Learning. PMLR, 2021.

**Audience:**

Yes

**Audience Explanation:**

I think the paper has made a decent first attempt to analyze a few interesting ideas in FG and ES which could inspire future investigations. Although the field of FG and ES is a relatively niche area, I still believe there are researchers who are interested in learning about the finds of this paper.

**Broader Impact Concerns:**

I don't have any concerns on ethical implications of this work.

**Claims And Evidence:**

Yes

**Claims Explanation:**

I think the claims in the submission are reasonably qualified, accurate, and supported by clear evidence.

**Requested Changes:**

- I think the paper should change the name Evolutionary Strategies to Evolution Strategies.
- I think the paper should double check all the captions mention all the symbols in the Figure.
- I think the paper could also make it more explicit what quadratic function each experiment is analyzing.
- I think the paper should make more explicit explanation of why they choose to only study the spherical quadratic. Maybe the authors can describe what difficulties they encounter for the more general case, and whether they have thoughts how to simply the assumption to make the analysis tractable.

---

> ### Author Response · Authors · 2026-05-01
>
> We thank the reviewer for their detailed and helpful feedback. We have made several changes to the paper to address the weaknesses and requested changes mentioned in the review, which we outline below. All figure numbers and equation numbers that we mention are referring to those in the new version of the PDF, uploaded on 30th April.
>
> - Connection between ES and FG: we respectfully disagree about this connection being known in the literature. Recall that we claim that *ES and FG are mathematically equivalent* for all *quadratic* objective functions.
>   - The MeZO paper in Definition 1 states that the ES + antithetic sampling gradient estimator *approximates* the expression that we give in Equation 16. They do not mention that this approximation is *exact* (or that it is exactly equal to the FG gradient estimator) for all quadratic objective functions.
>   - The PES paper proves that their estimator is unbiased for quadratic losses, but they do not mention or cite FG. This is related to the discussion we provide in the final paragraph of Section 3 about the unbiasedness of ES, and so we have added a reference to the PES paper in that paragraph. However, this does not take away from the central claim of Section 3, which is that ES and FG are mathematically equivalent for all quadratic objective functions.
> - Simplified quadratic setup: this was a common criticism from all reviewers. To address this criticism, we have added two new sections and a new figure:
>   - In section 4.3, we present theoretical results for smooth PL objective functions, which generalise our earlier theoretical results. We also reference existing work that justifies the smooth PL assumption as a model for analysing neural network optimisation.
>   - In section 4.4 we present empirical results comparing FG and ES against backpropagation for training neural networks. We added a new figure (Figure 2 in the new PDF) with the numerical results for this section. We include a discussion of how these empirical results relate to our other theoretical and empirical results, and also how they relate to results from Baydin et al. (2022).
>   - In section 8, we also added a new paragraph (the penultimate paragraph, beginning "Besides our theoretical and experimental results…") explaining in more detail our rationale for considering quadratic objective functions in sections 5, 6, and 7.
>   - The review mentioned that gradient descent converges to the global minimum of the spherical quadratic objective function in a single step of gradient descent. Recall that we make the same observation in the paragraph above Equation 23 in Section 4.2. This observation highlights that we should expect any optimiser to solve this optimisation problem easily, and our results demonstrating that FG and ES do *not* solve this optimisation problem easily highlight the weakness of FG and ES in high dimensions, which we believe is a valuable contribution.
> - Momentum methods:
>   - The observation in the review that gradient descent with momentum can also solve the spherical quadratic objective function in a single step by setting the momentum to zero is insightful, and in fact analogous to our results in Section 4.2. We have added a statement at the end of the paragraph below Equation 38 in Section 5 highlighting this observation.
>   - "it is hard to know whether this is more a property of the exact spherical quadratic setup or a genuine issue with momentum + FG": our new results in Sections 4.3 and 4.4 demonstrate that this scalability problem is a genuine fundamental issue with FG, and is not a property of the exact spherical quadratic setup. As we discuss in the second paragraph of Section 5, there may be two motivations for using momentum: (1) averaging out noise in the FG gradient estimates, and (2) averaging out fluctuations in the gradient direction that occur when the condition number of the Hessian is large. As we mention in the last paragraph of Section 5, our reason for studying the spherical quadratic is to deconfound these 2 factors, and our results show that momentum does not help to address the first of these 2 factors. In general, we would expect momentum to help with the second of these 2 factors, as it does for exact gradient descent.
>
> (continued in next comment)

---

> > ### Author Response · Authors · 2026-05-01
> >
> > (continued from previous comment)
> >
> > - Running actual experiments: We have added experimental results for neural network optimisation in Section 4.4, with numerical results in Figure 2. Additionally, Figure 1, Figure 3 (right), Figure 4 (right), and Figure 7 all contain experimental results from numerical simulations on the spherical quadratic objective function. Consistent with Figure 1, we have added theoretical predictions to Figure 3 (right) and Figure 7, to validate our theoretical predictions, and to distinguish experimental results from theoretical predictions. These figures show that the *asymptotic* convergence rates that we derive are largely consistent with our experimental results.
> > - Specifying objective functions: We have added Equations 37 and 38 explicitly defining the spherical quadratic objective function, and updated every subsequent section and figure caption to mention that they are applied to the spherical quadratic objective function.
> > - Dashed lines in control variate plot (previously Figure 3, now Figure 4): we have updated the caption of figure 4 to explain the meaning of the dashed lines.
> > - Correct name of Evolution Strategies: we have updated the paper, title, and abstract to correct "Evolutionary Strategies" to "Evolution Strategies".
> > - "explicit explanation of why they choose to only study the spherical quadratic": as mentioned above, we have broadened the scope of the paper by adding new sections (4.3 and 4.4) and a new figure (Figure 2) theoretically and experimentally considering smooth PL objective functions and neural network optimisation respectively, and we have added a new paragraph in Section 8 (the penultimate paragraph, beginning "Besides our theoretical and experimental results…") explaining in more detail our rationale for considering quadratic objective functions in Sections 5, 6, and 7.
> >
> > We once again thank the reviewer for their detailed and helpful review. We hope these changes and explanations address the reviewer’s concerns.

---

### Review · Reviewer_qFJZ · 2026-04-22

**Summary Of Contributions:**

The paper studies stochastic gradient estimators, revealing a mathematical equivalence between Forward Gradients (FG) and a variant of Evolutionary Strategies (ES) on quadratic objectives. The authors also analyze the scalability of both methods and attempt to improve the performance by using variance reduction.

Strengths
1. The studied problem is meaningful and interesting.
2. The paper is mostly clear and easy to follow.


Weaknesses

My main concern is that the theoretical analysis is restricted to quadratic cases, which significantly limits the contribution of this paper.

**Audience:**

No

**Audience Explanation:**

FG and ES are typically used to compute gradients for neural networks, which are highly nonconvex. However, the theoretical analysis in this paper focuses only on quadratic problems. As a result, its findings may have limited appeal to the broader TMLR audience.

**Claims And Evidence:**

No

**Claims Explanation:**

The theoretical analysis is limited to quadratic cases, and it remains unclear whether the proposed theory extends to general convex or nonconvex problems.

**Requested Changes:**

I think the authors should extend their theoretical analysis to general convex and nonconvex problems.

---

> ### Author Response · Authors · 2026-04-30
>
> We thank the reviewer for their constructive feedback. The primary concern, which is that we only present theoretical and experimental results for *quadratic* objective functions, without considering general convex or nonconvex problems, was a common criticism from all reviewers. To address this criticism, we have added two new sections and a new figure, which we summarise below.
>
> In section 4.3, we present theoretical results for smooth PL objective functions, which generalise our earlier theoretical results. We also reference and discuss existing work that justifies the smooth PL assumption as a model for analysing neural network optimisation. The PL assumption is quite general, including nonconvex objective functions (see references in section 4.3, paragraph 1, in the updated version of the paper).
>
> In section 4.4 we present empirical results comparing FG and ES against backpropagation for training neural networks. We added a new figure (Figure 2 in the new PDF) with the numerical results for this section. We include a discussion of how these empirical results relate to our other theoretical and empirical results, and also how they relate to results from Baydin et al. (2022).
>
> In section 8, we added a new paragraph (the penultimate paragraph, beginning “Besides our theoretical and experimental results…”) explaining in more detail our rationale for considering quadratic objective functions in sections 5, 6, and 7.
>
> We once again thank the reviewer for their constructive feedback. We hope these changes and explanations address the reviewer’s concerns.

---

> > ### Author Response · Authors · 2026-05-06
> >
> > Before the end of the rebuttal period, we would like to make one further point. The review answered "No" to the question "Are the claims made in the submission supported by accurate, convincing and clear evidence?", because:
> >
> > > The theoretical analysis is limited to quadratic cases, and it remains unclear whether the proposed theory extends to general convex or nonconvex problems.
> >
> > If, in the original paper we had *claimed* that the proposed theory extends to general convex or nonconvex problems, then we would agree that the answer to this question should be "No". However, we did not make this claim, and we would argue that the claims that we *did* make *were* supported by accurate, convincing and clear evidence. Therefore we would argue that the answer to this question is "Yes".
> >
> > Anyhow, as previously mentioned, we have now added more theory and experiments considering nonconvex problems, including neural network optimisation. We hope these changes and explanations address the reviewer’s concerns, and we once again thank the reviewer for their constructive feedback.

---

### Author Response · Authors · 2026-06-23

We have uploaded the deanonymised camera ready version of the paper.

We would like to take the opportunity to once again thank the Action Editor and reviewers for facilitating the review process and providing constructive feedback that helped to improve the paper!

---

### Decision · Action_Editor_rr5o · 2026-06-17

**Recommendation:** Accept as is

**Audience:**

Yes

**Audience Explanation:**

The theoretical analysis of the relationship between FG and ES, alongside the insights into using control variates in FG, will be of interest to researchers working in these fields. Although the initial analysis was restricted to quadratic objective functions, the authors successfully extended their framework to a more general class of smooth PL functions and provided supporting experiments on neural network optimization.

**Claims And Evidence:**

Yes

**Claims Explanation:**

This paper provides rigorous proof of the equivalence of FG and ES on quadratic objective functions. Their claims about the role the momentum and the use of unbiased control variate are supported by the experiments.